# BEYOND THE CONVEXITY ASSUMPTION: REALISTIC TABULAR DATA GENERATION UNDER QUANTIFIER-FREE REAL LINEAR CONSTRAINTS

**Mihaela Cătălina Stoian**
University of Oxford
mihaela.stoian@cs.ox.ac.uk

**Eleonora Giunchiglia**
Imperial College London
e.giunchiglia@imperial.ac.uk

## ABSTRACT

Synthetic tabular data generation has traditionally been a challenging problem due to the high complexity of the underlying distributions that characterise this type of data. Despite recent advances in deep generative models (DGMs), existing methods often fail to produce realistic datapoints that are well-aligned with available background knowledge. In this paper, we address this limitation by introducing Disjunctive Refinement Layer (DRL), a novel layer designed to enforce the alignment of generated data with the background knowledge specified in user-defined constraints. DRL is the first method able to automatically make deep learning models inherently compliant with constraints as expressive as quantifier-free linear formulas, which can define non-convex and even disconnected spaces. Our experimental analysis shows that DRL not only guarantees constraint satisfaction but also improves efficacy in downstream tasks. Notably, when applied to DGMs that frequently violate constraints, DRL eliminates violations entirely. Further, it improves performance metrics by up to 21.4% in F1-score and 20.9% in Area Under the ROC Curve, thus demonstrating its practical impact on data generation.

## 1 INTRODUCTION

The problem of tabular data generation is a critical area of research, driven by its numerous practical applications across various domains. High-quality synthetic data offers solutions to pressing challenges such as data scarcity (Choi et al., 2017), bias in unbalanced datasets (van Breugel et al., 2021), and the general need for privacy protection (Lee et al., 2021). However, due to the varied nature of the data distributions in the tabular domain—which are often multi-modal, and present complex dependencies among features—it is difficult to create models able to generate realistic data. Indeed, no matter the Deep Generative Model (DGM) used, when synthetic datapoints are tested for alignment with the available background knowledge, they frequently fail such a test. Even when considering simple knowledge like *"the feature representing the maximum recorded level of hemoglobin should be greater than or equal to the one representing its minimum"*, DGMs often generate datapoints violating it. So far, this problem has only been solved by either rejecting the non-aligned samples, or by adding a layer to the DGM that restricts its output space to coincide with the one defined by linear inequalities (Stoian et al., 2024). However, while the first solution is not feasible in the presence of a high violation rate, the second is only available when the knowledge can be captured by linear inequalities, which have very limited expressivity.

In this paper, we propose a novel layer—called Disjunctive Refinement Layer (DRL)—able to constrain any DGM output space according to background knowledge expressed as Quantifier-Free Linear Real Arithmetic (QFLRA) formulas. QFLRA formulas can capture any relationship over the features that can be represented as a combination of conjunctions, disjunctions and negations of linear inequalities. Thanks to their expressivity, QFLRA formulas can define spaces that are not only non-convex but can also be disconnected. On the contrary, linear inequalities can only capture convex output spaces. See Figure 1 for an example of spaces defined by linear

Figure 1: Example of spaces defined by (a) a set of linear inequalities and (b) a set of QFLRA formulas.

inequalities and QFLRA formulas. While linear inequalities establish a single lower and upper bound (if existent) for each feature, QFLRA formulas define multiple intervals where the background knowledge holds, each with its own boundaries. This significantly increases the complexity of the problem, as compiling knowledge into DRL not only requires keeping track of these intervals but also deriving the intricate hidden interactions among variables.

**Example 1.** *The knowledge: "The value of $x_5$ should be always at least $x_1$, and if greater than $x_2$ then it should also be at least equal to $x_3$. In any case, $x_5$ should never be greater than $x_4$", which cannot be expressed by a set of linear inequalities, corresponds to the QFLRA formula:*

$$(x_5 \geq x_1) \wedge ((x_5 > x_2) \rightarrow (x_5 \geq x_3)) \wedge (x_5 \leq x_4). \tag{1}$$

*Moreover, this formula entails other hidden relations among the variables such as, e.g., $\neg(x_1 > x_4)$.*

To derive such additional hidden relations, we developed a novel variable elimination method which generalises the analogous procedure for systems of linear inequalities based on the Fourier-Motzkin result (see, e.g., (Dechter, 1999)). Once compiled, by definition, DRL (i) guarantees the satisfaction of the constraints, (ii) can be seamlessly added to the topology of any neural model, (iii) allows the backpropagation of the gradients at training time, (iv) performs all the computations in a single forward pass (i.e., no cycles), and (v) given a sample generated by a DGM, it returns a new one that is optimal with respect to the original (intuitively, which minimally differs from the original sample while taking into account the user preferences on which features should be changed first). Our experimental analysis also shows that adding DRL to DGMs improves their machine learning efficacy (Xu et al., 2019) on a range of different scenarios. This is the most widely used performance measure for evaluating the quality of synthetic data, as it assesses how useful the generated data is for downstream tasks. In particular, we considered five DGMs, added DRL into their topology and got improvements for all datasets of up to 21.4%, 20.5%, and 20.9% in terms of F1-score, weighted F1-score, and Area Under the ROC Curve, respectively. Finally, our experiments demonstrate a strong need for a method like ours. Indeed, DGMs generate synthetic datapoints violating the background knowledge more often than expected. In 13 out of 25 scenarios, the DGMs produced datasets with over 50% datapoints violating the constraints, and in five cases this reached 100%.

**Main contributions:** (i) We propose the first-ever layer that can be integrated into any DGM to enforce background knowledge expressed as QFLRA formulas. This required generalising the Fourier-Motzkin variable elimination procedure in order to handle disjunctions of linear inequalities. (ii) We show experimentally how integrating our layer in DGMs improves their machine learning efficacy, even when the constraints define a possibly non-convex and disconnected space.

## 2 PROBLEM DEFINITION AND NOTATION

*Constrained generative modelling* is defined as the problem of learning the parameters $\theta$ of a generative model, given an unknown distribution $p_X$ over $X \in \mathbb{R}^D$, a training dataset $\mathcal{D}$ consisting of $N$ i.i.d. samples drawn from $p_X$, and formally expressed background knowledge about the problem—stating which samples are admissible and which are not—such that (i) the model distribution $p_\theta$ approximates $p_X$, and (ii) the sample space of $p_\theta$ is aligned with what is stated in the background knowledge. As described in the introduction, so far this problem has only been solved by either rejecting the non-aligned samples or by including a layer into the DGM that restricts its output space to coincide with the one defined by the linear inequalities (Stoian et al., 2024).

In this paper, we allow for background knowledge expressed as a set of formulas, each being a disjunction of linear inequalities. This enables us to capture any relationship among features which can be represented as Quantifier-Free Linear Real Arithmetic (QFLRA) formulas. Indeed, through syntactic manipulation using De Morgan's laws and the mathematical properties of linear inequalities, any knowledge formulated as a combination of conjunctions, disjunctions, and negations of linear inequalities can be rewritten as a set of disjunctions over linear inequalities. Formally, we consider a set $\Pi$ of constraints, where a *constraint* is a disjunction of $n_\Psi \in \mathbb{N}$ linear inequalities of the form:

$$\Psi = \Phi_1 \vee \Phi_2 \vee \cdots \vee \Phi_{n_\Psi}, \tag{2}$$

where each $\Phi_i$ is a linear inequality over the set of variables $\mathcal{X} = \{x_k \mid k = 1, \ldots, D\}$, each variable uniquely corresponding to a feature in the dataset. We assume each linear inequality has form:

$$\sum_k w_k x_k + b \geq 0, \tag{3}$$

with $w_k \in \mathbb{R}$, $b \in \mathbb{R}$, and $x_k$ ranging over $\mathbb{R}$. When $w_i \neq 0$, we say that $x_i$ *occurs* in (3), and that it occurs *positively* if $w_i > 0$ and *negatively* otherwise.

For an easy formulation of the problem and its solution, given two linear expressions $\varphi = \sum_k w_k x_k + b$ and $\varphi' = \sum_k w'_k x_k + b'$, we write, e.g., $(\varphi + \varphi')$ for the linear expression $\sum_k (w_k + w'_k) x_k + (b + b')$, and similarly for $(\varphi - \varphi')$ and $\varphi/w$ if $w \in \mathbb{R} \setminus \{0\}$. We will also express the linear inequality (3) as $w_i x_i + \varphi$, by this implicitly assuming $w_i \neq 0$ and that $x_i$ does not occur in $\varphi$, i.e., that $\varphi = \sum_{k \neq i} w_k x_k + b$. Finally, we also write $\varphi \geq \varphi'$ (resp. $\varphi \leq \varphi'$) as abbreviations for $\varphi - \varphi' \geq 0$ (resp. $\varphi' - \varphi \geq 0$).

Given a DGM with distribution $p_\theta$, a *sample* $\tilde{x} \sim p_\theta$ is an assignment to the variables in $\mathcal{X}$, and $\tilde{x}_k$ indicates the value assigned by $\tilde{x}$ to the variable $x_k$. We say that a sample $\tilde{x}$ *satisfies*

- the linear inequality (3) if $\sum_k w_k \tilde{x}_k + b \geq 0$,
- the constraint $\Psi$ with form (2) if $\Phi_i$ is satisfied by $\tilde{x}$ for some $i = 1, \ldots, n_\Psi$, and
- a set $\Pi$ of constraints if $\tilde{x}$ satisfies all the constraints in $\Pi$.

Further, we associate to each linear inequality $\Phi$, (resp. constraint $\Psi$, resp. set of constraints $\Pi$) the set $\Omega(\Phi)$ (resp. $\Omega(\Psi)$, resp. $\Omega(\Pi)$) of the points in $\mathbb{R}^D$ that satisfy $\Phi$ (resp. $\Psi$, resp. $\Pi$). Clearly, $\Omega(\Phi)$, $\Omega(\Psi)$ and $\Omega(\Pi)$ define a subspace of $\mathbb{R}^D$, and have the following properties:

1. $\Omega(\Phi)$ is non-empty and convex,
2. $\Omega(\Psi)$ is non-empty but may be non-convex and also disconnected, and
3. $\Omega(\Pi)$ may be empty, non-convex and also disconnected,

all the above assuming some variable occurs in $\Phi$, $\Psi$ and $\Pi$. A linear inequality $\Phi$ (resp. a constraint $\Psi$, resp. a set of constraints $\Pi$) is *violated* by a sample $\tilde{x}$ if $\tilde{x}$ does not belong to the corresponding set $\Omega(\Phi)$ (resp. $\Omega(\Psi)$, resp. $\Omega(\Pi)$). A linear inequality $\Phi$ (resp. a constraint $\Psi$, resp. a set of constraints $\Pi$) is *satisfiable* if the corresponding set $\Omega(\Phi)$ (resp. $\Omega(\Psi)$, resp. $\Omega(\Pi)$) is not empty. Notice that, for the sake of simplicity, we do not consider strict inequalities (i.e., inequalities with $>$). From a theoretical perspective, the entire theory can be easily generalised to consider them. From a practical perspective, for any computing system of choice, we can simply rewrite each strict inequality of the following form: $\sum_k w_k x_k + b > 0$ as $\sum_k w_k x_k + b - \epsilon \geq 0$, where $\epsilon > 0$ denotes the desired precision of the representation, taking into account the limitations of floating-point accuracy. Finally, we represent each constraint $\Psi \in \Pi$ of form (2) also as the set $\{\Phi_1, \Phi_2, \ldots, \Phi_{n_\Psi}\}$. With this notation, $\Pi$ is a set of sets of linear inequalities. Hence, the linear inequalities in a set should be interpreted as disjunctively defining a constraint in $\Pi$, while the constraints are to be interpreted as conjunctively defining $\Pi$.

## 3 DISJUNCTIVE REFINEMENT LAYER

Given a finite set of constraints $\Pi$ and a DGM, we show how to build a layer with all the desired properties stated in the introduction. In Appendix A we visualize how to add our DRL to each of the DGMs considered in the experimental analysis. Before illustrating the general case, in the following subsection we assume $\Pi$ is a finite set of constraints in a single variable $x_i$.

### 3.1 SINGLE VARIABLE CASE

Each constraint $\Psi$ of form (2) defines a single *left boundary* $l_i^\Psi$ and a single *right boundary* $r_i^\Psi$ for the variable $x_i$: [1]

$$l_i^\Psi = \max_{(w_i x_i + \varphi \geq 0) \in \Psi : w_i < 0} \left( - \frac{\varphi}{w_i} \right), \qquad r_i^\Psi = \min_{(w_i x_i + \varphi \geq 0) \in \Psi : w_i > 0} \left( - \frac{\varphi}{w_i} \right). \qquad (4)$$

Assuming $\Psi$ contains a linear inequality in which $w_i \neq 0$, a sample $\tilde{x}$ satisfies $\Psi$ if and only if either $\tilde{x}_i \leq l_i^\Psi$ or $\tilde{x}_i \geq r_i^\Psi$, as represented in Figure 2. As the Figure clearly shows, $\Omega(\Psi)$ is already non-convex whenever $l_i^\Psi \neq -\infty$, $r_i^\Psi \neq +\infty$ and $l_i^\Psi < r_i^\Psi$. When considering a set $\Pi$ with multiple

---

[1]We use the "left" and "right" terminology because in a non-vacuous constraint we have $l_i^\Psi < r_i^\Psi$. We assume the function $\min(\mathcal{S})$ over a finite set $\mathcal{S}$ of values in $\mathbb{R}$ to be defined as $\min(\emptyset) = +\infty$, and $\min(\{v\} \cup \mathcal{S}') = v$ if $v \leq \min(\mathcal{S}')$ and $\min(\mathcal{S}')$ otherwise. Analogously for the function $\max(\mathcal{S})$.

constraints in $x_i$, we may arrive at a set $\Omega(\Pi)$ that is the union of up to $|\Pi| + 1$ disjoint intervals.

In general, computing the intervals requires finding the satisfying boundaries and then ordering them. Luckily, given a sample $\tilde{x}$ violating some constraint in $\Pi$, we are only interested in setting $\mathrm{DRL}(\tilde{x})_i$ equal to the bound that satisfies $\Pi$ and that is at minimal Euclidean distance from $\tilde{x}_i$. To this end, we first define the *closest satisfying left and right boundary for $\tilde{x}_i$* as:

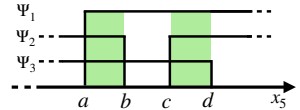

Figure 2: Visualisation of left and right boundaries defined by constraint $\Psi$. The green regions correspond to the values of $x_i \in \Omega(\Psi)$.

$$l_i^{\Pi}(\tilde{x}) = \max_{\Psi \in \Pi}(\{l_i^{\Psi} : \tilde{x}_i > l_i^{\Psi}, l_i^{\Psi} \in \Omega(\Pi)\}), \quad r_i^{\Pi}(\tilde{x}) = \min_{\Psi \in \Pi}(\{r_i^{\Psi} : \tilde{x}_i < r_i^{\Psi}, r_i^{\Psi} \in \Omega(\Pi)\}), \quad (5)$$

respectively. Then, for $k \neq i$, $\mathrm{DRL}(\tilde{x})_k = \tilde{x}_k$ and

$$\mathrm{DRL}(\tilde{x})_i = \begin{cases} \tilde{x}_i & \text{if } \tilde{x} \in \Omega(\Pi), \\ l_i^{\Pi}(\tilde{x}) & \text{if } \tilde{x} \notin \Omega(\Pi) \text{ and } |\tilde{x}_i - l_i^{\Pi}(\tilde{x})| < |\tilde{x}_i - r_i^{\Pi}(\tilde{x})|, \\ r_i^{\Pi}(\tilde{x}) & \text{otherwise.} \end{cases} \quad (6)$$

By construction, $\mathrm{DRL}(\tilde{x})$ satisfies the constraints in $\Pi$ and is *optimal w.r.t. $\tilde{x}$*: there does not exist a sample satisfying $\Pi$ with smaller Euclidean distance from $\tilde{x}$.

**Lemma 3.1.** *Let $\Pi$ be a finite and satisfiable set of constraints in a single variable $x_i$. For every sample $\tilde{x}$, $\mathrm{DRL}(\tilde{x})$ satisfies $\Pi$ and is optimal w.r.t. $\tilde{x}$.*

The proof of the Lemma can be found in Appendix B.

**Example 2.** *Let $\Pi$ be the set of constraints $\{\Psi_1, \Psi_2, \Psi_3\}$ over the unique variable $x_5$, with $\Psi_1, \Psi_2$ and $\Psi_3$ as shown in Figure 3. Then, $l_5^{\Psi_1} = -\infty, l_5^{\Psi_2} = b, l_5^{\Psi_3} = d, r_5^{\Psi_1} = a, r_5^{\Psi_2} = c, r_5^{\Psi_3} = +\infty$. Depending on the value of $\tilde{x}_5$, we get correspondingly different values for $\mathrm{DRL}(\tilde{x})_5$. In particular,*

Figure 3: Constraints for Ex. 2.

1. *if $\tilde{x}_5 < a$ then $\mathrm{DRL}(\tilde{x})_5 = a$,*
2. *if $a \leq \tilde{x}_5 \leq b$ then $\mathrm{DRL}(\tilde{x})_5 = \tilde{x}_5$,*
3. *if $b < \tilde{x}_5 < (b+c)/2$ then $\mathrm{DRL}(\tilde{x})_5 = b$,*
4. *if $(b+c)/2 \leq \tilde{x}_5 < c$ then $\mathrm{DRL}(\tilde{x})_5 = c$,*
5. *if $c \leq \tilde{x}_5 \leq d$ then $\mathrm{DRL}(\tilde{x})_5 = \tilde{x}_5$,*
6. *if $\tilde{x}_5 > d$ then $\mathrm{DRL}(\tilde{x})_5 = d$.*

*Independently from the value of $\tilde{x}_5$, $\mathrm{DRL}(\tilde{x})_5$ satisfies the constraints and is optimal w.r.t. $\mathrm{DRL}(\tilde{x})_5$.*

## 3.2 GENERAL CASE

Among the desiderata for our DRL, we have that all the necessary computations need to be done in a single forward pass. To this end, we consider a *variable ordering* $x_1; x_2; \ldots; x_D$ corresponding to the order of computation of the features. The ordering can be arbitrarily selected or, more appropriately, may reflect the user preferences on which features should be changed first when the sample violates the constraints. Indeed, the value of each feature $x_i$ will be computed taking into account the values of the features $x_1, \ldots, x_{i-1}$, the latter considered immutable. To make this possible, when building the layer, we need to ensure that the chosen value for the variables $x_j$, with $j < i$, guarantees the existence of a value for $x_i$ satisfying the constraints. Starting from $\Pi_D = \Pi$ and $i = D$, this amounts to deriving a finite set $\Pi_{i-1}$ of constraints in the variables $x_1, x_2, \ldots, x_{i-1}$ whose conjunction is logically equivalent to $\exists x_i \bigwedge_{\Psi \in \Pi_i} \Psi$. This entails that for every value of $x_1, x_2, \ldots, x_{i-1}$ satisfying $\Pi_{i-1}$ there must exist a value for $x_i$ satisfying $\Pi_i$, or alternatively, that each assignment to $x_1, x_2, \ldots, x_{i-1}$ and satisfying $\Pi_{i-1}$ can be extended to satisfy also $\Pi_i$.

In order to define such set $\Pi_{i-1}$, given two constraints $\Psi = (\bigvee_{k=1}^{n}(w_k x_i + \varphi_k \geq 0) \vee \Phi)$ and $\Psi' = (\bigvee_{j=1}^{m}(w'_j x_i + \varphi'_j \geq 0) \vee \Phi')$, with $w'_1, \ldots, w'_m < 0 < w_1, \ldots, w_n$ and $m, n \geq 1$, we define the *cutting planes (CP) resolution rule* between $\Psi$ and $\Psi'$ on $x_i$ to be:

$$\frac{\bigvee_{j=1}^{m}(w'_j x_i + \varphi'_j \geq 0) \vee \Phi' \qquad \bigvee_{k=1}^{n}(w_k x_i + \varphi_k \geq 0) \vee \Phi}{\bigvee_{j=1}^{m} \bigvee_{k=1}^{n}(\varphi_k/w_k - \varphi'_j/w'_j \geq 0) \vee \Phi \vee \Phi'}. \quad (7)$$

In the above rule, $\Psi$ and $\Psi'$ are the *premises*, and the formula below the line is the *conclusion* denoted with $CPres_i(\Psi, \Psi')$. This rule, which can be derived from the standard propositional and

CP rules defined, e.g., in (Krajícek, 1998), is sound for any possible $\Phi$ and $\Phi'$. Despite this, we assume that $\Phi'$ (resp. $\Phi$) does not contain negative (resp. positive) occurrences of $x_i$. As we will see, it is possible to impose much stronger conditions (defined later) on the applicability of the rule, still enabling the derivation of a set of constraints $\Pi_{i-1}$ with the desired properties.

**Lemma 3.2.** *The CP resolution rule is sound: the premises entail the conclusion of the rule.*

**Example 3** (Example 1, cont'd). *The QFLRA formula in the introduction translates into the set of constraints:* $\Pi = \{\Psi_1, \Psi_2, \Psi_3\}$ *with* $\Psi_1 = (x_5 \geq x_1), \Psi_2 = ((x_5 \leq x_2) \vee (x_5 \geq x_3))$ *and* $\Psi_3 = (x_5 \leq x_4)$. *By applying the CP resolution rule, we can obtain a new set of constraints entailed by* $\Pi$ *and logically equivalent to* $\exists x_5 \bigwedge_{\Psi \in \Pi} \Psi$.

$$CPres_5(\Psi_1, \Psi_2) = x_1 \leq x_2 \vee x_5 \geq x_3 \qquad CPres_5(\Psi_1, \Psi_3) = x_1 \leq x_4$$

$$CPres_5(\Psi_3, CPres_5(\Psi_1, \Psi_2)) = x_1 \leq x_2 \vee x_3 \leq x_4.$$

*As derived from the multiple application of the CP resolution rule, the above set of constraints admits a solution for* $x_5$ *if and only if* $(x_1 \leq x_4) \wedge (x_1 \leq x_2 \vee x_3 \leq x_4)$.

The proof of Lemma 3.2 is in Appendix C. In the example, there is only one constraint with both positive and negative occurrences of $x_i$ and the CP resolution of any two distinct constraints always leads to a conclusion with either only positive or negative occurrences of $x_i$. However, in general, the CP resolution of two constraints $\Psi$ and $\Psi'$ will lead to a new constraint $CPres_i(\Psi, \Psi')$ which might contain both positive and negative occurrences of $x_i$. This new constraint can be the premise of other CP resolutions which can produce new constraints and the process can iterate. Nevertheless, our goal is to derive the constraints in the variables $x_1, \ldots, x_{i-1}$ whose satisfying assignments can be extended to satisfy also the constraints with $x_i$. The standard solution to make all the possible CP resolutions on $x_i$ while considering also the CP resolvent of the already done resolution may turn out to be too computationally expensive. Luckily, we can further restrict to CP resolutions between two constraints $\Psi$ and $\Psi'$ in which $\Psi$ does not contain negative occurrences of $x_i$. To this end, let

1. $\Pi_i^+$ (resp. $\Pi_i^-$) to be the set of constraints in $\Pi_i$ with (resp. without) positive occurrences of $x_i$ and without (resp. with) negative occurrences of $x_i$ ;

2. $\Pi_i^{\pm}$ to be the set of constraints in $\Pi_i$ with both positive and negative occurrences of $x_i$;

3. $\Pi_i^{\ddagger}$ to be the set of constraints obtained by the recursive application of the CP-resolution between one constraint without negative occurrences of $x_i$ and one constraint in $\Pi_i^{\pm}$:

$$\Pi_i^{\ddagger} = \bigcup_{k=0}^{|\Pi_i^{\pm}|} \Pi_i^k \qquad \text{with } \Pi_i^{k+1} = \{CPres_i(\Psi, \Psi') \mid \Psi \in \Pi_i^k, \Psi' \in \Pi_i^{\pm}\},$$

and $\Pi_i^0 = \Pi_i^+$. Every constraint in $\Pi_i^{\ddagger}$ has only positive occurrences of $x_i$.

Then, $\Pi_{i-1}$ is the set of constraints in $\Pi_i$ in which $x_i$ does not occur plus the set of constraints obtained by the CP resolution of the constraints in $\Pi_i^+ \cup \Pi_i^- \cup \Pi_i^{\ddagger}$. More formally,

$$\Pi_{i-1} = (\Pi_i \setminus (\Pi_i^+ \cup \Pi_i^- \cup \Pi_i^{\pm})) \cup \{CPres_i(\Psi, \Psi') \mid \Psi \in \Pi_i^{\ddagger}, \Psi' \in \Pi_i^- \}. \tag{8}$$

Clearly, each set $\Pi_{i-1}$ does not contain any occurrence of $x_i$ and can contain a non-polynomial number of constraints, the latter fact echoing similar results for variable elimination methods in propositional logic and sets of linear inequalities (Dechter, 1999). The above definition, generalises to disjunctions of linear inequalities the standard variable elimination procedure proposes for systems of linear inequalities based on the Fourier-Motzkin result.

**Example 4** (Example 3, cont'd). *Consider the variable ordering* $x_1; x_2; x_3; x_4; x_5$. *Then,* $\Pi_5 = \Pi, \Pi_5^- = \{\Psi_3\}, \Pi_5^{\pm} = \{\Psi_2\}, \Pi_5^+ = \Pi_5^0 = \{\Psi_1\}, \Pi_5^1 = \{x_1 \leq x_2 \vee x_5 \geq x_3\}$, *and* $\Pi_5^{\ddagger} = \Pi_5^0 \cup \Pi_5^1$. *As a consequence,* $\Pi_4 = \{x_1 \leq x_4, x_1 \leq x_2 \vee x_3 \leq x_4\}$, *and* $\Pi_3 = \Pi_2 = \Pi_1 = \emptyset$.

For each set of constraints $\Pi_i$, the set $\Pi_{i-1}$ has the desired property, stated in the lemma below.

**Lemma 3.3.** *Let* $\Pi$ *be a set of constraints in the variables* $x_1, \ldots, x_i$. $\Pi_i = \Pi$ *and* $\Pi_{i-1}$ *are equisatisfiable, and each assignment to the variables* $x_1, \ldots, x_{i-1}$ *satisfying* $\Pi_{i-1}$ *can be extended in order to satisfy* $\Pi_i$.

---

**Algorithm 1** Compile & Apply DRL

---

    **function** DRL_COMPILE($\Pi, x_1; \dots; x_D$)         **function** DRL_APPLY($\tilde{x}, \Pi_1, \dots, \Pi_D$)

        $\Pi_D \leftarrow \Pi$                      **for** $i \leftarrow 1$ **to** $D$ **do**

        **for** $i \leftarrow D$ **downto** 1 **do**            compute $\widetilde{\Pi}_i, \Omega(\widetilde{\Pi}_i), l_i^{\widetilde{\Pi}_i}(\tilde{x}), r_i^{\widetilde{\Pi}_i}(\tilde{x})$

           compute $\Pi_i^+, \Pi_i^-, \Pi_i^\pm, \Pi_i^\ddagger$           **if** $\tilde{x}_i \in \Omega(\widetilde{\Pi}_i)$ **then** DRL$(\tilde{x})_i \leftarrow \tilde{x}_i$

           $\Pi_{i-1} \leftarrow (\Pi_i \setminus (\Pi_i^+ \cup \Pi_i^- \cup \Pi_i^\pm)) \cup$       **else if** $|\tilde{x}_i - l_i^{\widetilde{\Pi}_i}(\tilde{x})| < |\tilde{x}_i - r_i^{\widetilde{\Pi}_i}(\tilde{x})|$ **then**

              $\{CPres_i(\Psi, \Psi') \,|\, \Psi \in \Pi_i^\ddagger, \Psi' \in \Pi_i^-\}$             DRL$(\tilde{x})_i \leftarrow l_i^{\widetilde{\Pi}_i}(\tilde{x})$

        **if** $\Pi_0$ is unsatisfiable **then**           **else**    DRL$(\tilde{x})_i \leftarrow r_i^{\widetilde{\Pi}_i}(\tilde{x})$

           **return** UNSAT_FLAG          **return** DRL$(\tilde{x})_1; \dots; $DRL$(\tilde{x})_D$

        **else return** $\Pi_1; \dots; \Pi_D$

---

The proof of the Lemma is in Appendix D. As a corollary of the above lemma we have that the CP resolution is *refutationally complete*: if $\Pi$ is unsatisfiable then it is possible to derive a disjunction of linear inequalities in $\Pi_0$ in which each inequality (3) has $w_i = 0$ for $i = 1, \dots, D$ and $b < 0$ (otherwise, it is possible to incrementally define assignments satisfying $\Pi_1, \Pi_2, \dots, \Pi_D = \Pi$). Thus, at the end of the layer construction, we are able to automatically detect $\Pi$ unsatisfiability, returning a corresponding value in such case.

**Corollary 3.4.** *For any finite set of constraints, the CP resolution rule is refutationally complete.*

Starting from $i = 1$, the value of DRL$(\tilde{x})_i$ is computed considering the constraints in $\widetilde{\Pi}_i$, where $\widetilde{\Pi}_i$ is the set of constraints in the variable $x_i$ obtained by substituting the variables $x_1, x_2, \dots, x_{i-1}$ with DRL$(\tilde{x})_1,$ DRL$(\tilde{x})_2, \dots,$ DRL$(\tilde{x})_{i-1}$ in $\Pi_i$. As in the single variable case, assuming $\tilde{x}_i$ violates some constraint in $\widetilde{\Pi}_i$, we define the *closest satisfying left and right boundaries for $\tilde{x}_i$* as:

$$l_i^{\widetilde{\Pi}_i}(\tilde{x}) = \max_{\Psi \in \widetilde{\Pi}_i}(\{l_i^\Psi : \tilde{x}_i > l_i^\Psi, l_i^\Psi \in \Omega(\widetilde{\Pi}_i)\}), \quad r_i^{\widetilde{\Pi}_i}(\tilde{x}) = \min_{\Psi \in \widetilde{\Pi}_i}(\{r_i^\Psi : \tilde{x}_i < r_i^\Psi, r_i^\Psi \in \Omega(\widetilde{\Pi}_i)\}).$$

Then, for $j > i$, DRL$(\tilde{x})_j = \tilde{x}_j$, for $j < i$, DRL$(\tilde{x})_j = $DRL$(\tilde{x})_i$ and

$$\text{DRL}(\tilde{x})_i = \begin{cases} \tilde{x}_i & \text{if } \tilde{x}_i \in \Omega(\widetilde{\Pi}_i), \\ l_i^{\widetilde{\Pi}_i}(\tilde{x}) & \text{if } \tilde{x}_i \notin \Omega(\widetilde{\Pi}_i) \text{ and } |\tilde{x}_i - l_i^{\widetilde{\Pi}_i}(\tilde{x})| < |\tilde{x}_i - r_i^{\widetilde{\Pi}_i}(\tilde{x})|, \\ r_i^{\widetilde{\Pi}_i}(\tilde{x}) & \text{otherwise.} \end{cases} \tag{9}$$

A simple, non-optimised version of the algorithm is given in Algorithm 1. The compilation step happens only once before training, while the application step is performed for each sample.

**Example 5** (Examples 2, 4, cont'd)**.** *Consider a sample $\tilde{x}$ where $\tilde{x}_1 = a, \tilde{x}_2 = b, \tilde{x}_3 = c,$ and $\tilde{x}_4 = d$ (i.e., arranged as in Figure 3). Then, since $\Pi_3 = \Pi_2 = \Pi_1 = \emptyset$, DRL leaves the values unchanged for the features $x_1, x_2, x_3$ and $DRL(\tilde{x})_1 = a, DRL(\tilde{x})_2 = b, DRL(\tilde{x})_3 = c$. Regarding $\tilde{x}_4$, we know that $\widetilde{\Pi}_4 = \{x_4 \geq a, a \leq b \vee x_4 \geq c\}$ which reduces to $\{x_4 \geq a\}$, and, since it is satisfied, $DRL(\tilde{x}) = d$. Finally, $\widetilde{\Pi}_5 = \{x_5 \geq a, x_5 \leq b \vee x_5 \geq c, x_5 \leq d\}$ and the value of $DRL(\tilde{x})_5$ can be computed on the ground of $\tilde{x}_5$, as detailed in Example 2.*

**Theorem 3.5.** *Let $\Pi$ be a finite and satisfiable set of constraints. For any sample $\tilde{x}$ and variable ordering, the corresponding sample DRL$(\tilde{x})$ satisfies $\Pi$.*

Further, considering the variable ordering $x_1; x_2; \dots; x_D$, DRL$(\tilde{x})$ is *optimal w.r.t. $\tilde{x}$ and $\Pi$ and the variable ordering*: for each $i = 1, \dots, D$ there does not exist a sample $\tilde{x}' \in \Omega(\Pi)$ such that $|\tilde{x}_i - \tilde{x}'_i| < |\tilde{x}_i - \text{DRL}(\tilde{x})_i|$, and for all $j < i$, $\tilde{x}'_j = \text{DRL}(\tilde{x})_j$.

**Theorem 3.6.** *Let $\Pi$ be a finite and satisfiable set of constraints. For any sample $\tilde{x}$ and variable ordering, the corresponding sample DRL$(\tilde{x})$ is optimal w.r.t. $\tilde{x}$, $\Pi$ and the variable ordering.*

The proofs of Theorems 3.5 and 3.6 are in Appendix E and F, respectively.

## 4 EXPERIMENTAL ANALYSIS

To assess how DRL[2] performs in practice, we conduct the following studies. First, in Section 4.1, we investigate whether our layer improves the quality of the synthetic data generated by standard

---

[2]The code is available at `https://github.com/mihaela-stoian/DRL_DGM`.

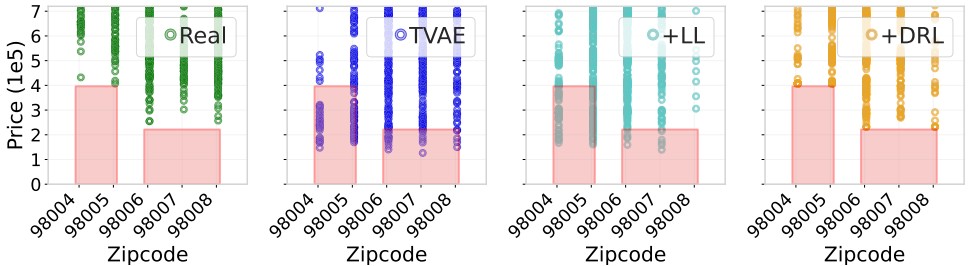

Figure 4: Sample distributions for real and synthetic data from TVAE, TVAE+LL and TVAE+DRL. The regions where samples violate the constraints are in red.

DGMs. In Section 4.2, we then compare our constrained models (which we refer to as DGMs+DRL) with the models obtained by considering only the linear constraints in each dataset and adding the layer proposed by Stoian et al. (2024). We refer to the linearly constrained DGMs as DGMs + Linear Layer (DGMs+LL). Then, in Section 4.3, we conduct experiments to determine how the background knowledge injection affects the sample generation time. Before delving into these studies, we describe the metrics we use to compute the sample quality, along with the models and datasets.

**Sample Quality Evaluation.** To judge the quality of our samples we measure (i) how well they align with the background knowledge and (ii) how well they can replace the real data in downstream tasks. To measure background knowledge alignment, we consider the metrics proposed in (Stoian et al., 2024): i,e, *constraint violation rate* (CVR), *constraint violation coverage* (CVC), and *samplewise constraints violation coverage* (sCVC). To determine their usability in downstream tasks we consider the metric *machine learning efficacy* (Kim et al., 2023), also known as utility (e.g., in (Liu et al., 2022)). To compute it, we follow the "Train on Synthetic, Test on Real" protocol (Esteban et al., 2017). Specifically, to compute the efficacy for classification (resp., regression) datasets, we train six classifiers (resp., four regressors) on synthetic data and test them on real data. A detailed description of the evaluation protocol and the hyperparameter tuning description for the classifiers and regressors can be found in Appendix I. For classification datasets, we report: F1-score (F1), weighted F1-score (*w*F1), and Area Under the ROC Curve (AUC), while for the regression dataset, we compute the mean absolute error (MAE) and the root mean square error (RMSE). For reference, we report the same metrics when training on the real data in Table 25 of Appendix O.

**Models.** We consider five DGMs: WGAN (Arjovsky et al., 2017), TableGAN (Park et al., 2018), CTGAN (Xu et al., 2019), TVAE (Xu et al., 2019), and GOGGLE (Liu et al., 2022), and build our DRL on top of each to create DGM+DRL models. A description of these models is in Appendix H.

**Datasets.** We consider five real-world datasets and associated constraints. Four datasets (i.e., URL, CCS, LCLD, and Heloc) are used for classification tasks, while one dataset (i.e., House) is used for regression. A detailed description of the datasets and their respective constraints are in Appendix G.

## 4.1 SYNTHETIC DATA QUALITY

**Background knowledge alignment.** To assess how often the samples violate the constraints, we calculate the CVR, which is defined as the percentage of samples that violate at least one constraint. Table 1 shows the CVR for each unconstrained model (first five rows) and our models equipped with the DRL (last row). More detailed findings are reported

Table 1: CVR for each model and dataset. Cases with CVR$\geq$50% are underlined. Best results are in bold.

|  | URL | CCS | LCLD | Heloc | House |
|---|---|---|---|---|---|
| WGAN | 22.8±4.9 | 44.7±7.1 | 47.5±14.5 | 80.6±9.3 | 100.0±0.0 |
| TableGAN | 8.5±2.2 | 61.2±13.3 | 32.0±4.7 | 59.9±16.7 | 100.0±0.0 |
| CTGAN | 9.7±2.0 | 78.5±5.7 | 7.1±1.3 | 56.6±9.8 | 100.0±0.0 |
| TVAE | 10.3±1.1 | 16.9±1.6 | 10.3±0.6 | 44.9±1.0 | 100.0±0.0 |
| GOGGLE | 7.3±8.1 | 60.3±6.8 | 70.4±16.1 | 52.7±6.3 | 100.0±0.0 |
| All + DRL | **0.0**±0.0 | **0.0**±0.0 | **0.0**±0.0 | **0.0**±0.0 | **0.0**±0.0 |

in Appendix K (Table 8), where the results for sCVC and CVC can also be found (Tables 9, 10). As expected, the models with our DRL always satisfy the constraints, while the samples obtained with standard DGMs very often violate them. Additionally, in many cases, the CVR is extremely high: out of 25 cases, 13 cases have CVR greater than 50% and 5 cases have CVR equal to 100%, thus

Table 2: Efficacy comparison between the unconstrained DGMs, and their +DRL and +RS counterparts. The performance is measured using F1, $w$F1, and AUC, for each classification dataset.

| | F1 | | | | wF1 | | | | AUC | | | |
|---|---|---|---|---|---|---|---|---|---|---|---|---|
| | URL | CCS | LCLD | Heloc | URL | CCS | LCLD | Heloc | URL | CCS | LCLD | Heloc |
| WGAN | 0.794 | 0.303 | 0.139 | 0.665 | 0.796 | 0.330 | 0.296 | 0.648 | 0.870 | 0.814 | 0.605 | **0.717** |
| + RS | 0.792 | 0.051 | 0.156 | 0.628 | 0.794 | 0.088 | 0.312 | 0.617 | 0.862 | 0.570 | 0.611 | 0.685 |
| + DRL | **0.800** | **0.313** | **0.197** | **0.721** | **0.801** | **0.340** | **0.339** | **0.652** | **0.875** | **0.885** | **0.623** | **0.717** |
| TableGAN | 0.562 | **0.196** | 0.259 | 0.593 | 0.659 | **0.228** | 0.393 | 0.615 | 0.843 | **0.802** | 0.655 | 0.707 |
| + RS | 0.544 | 0.138 | 0.251 | 0.568 | 0.648 | 0.172 | 0.389 | 0.599 | 0.854 | 0.682 | 0.653 | 0.685 |
| + DRL | **0.619** | 0.163 | **0.269** | **0.628** | **0.693** | 0.196 | **0.401** | **0.628** | **0.865** | 0.742 | **0.657** | **0.709** |
| CTGAN | 0.822 | 0.145 | 0.247 | 0.736 | 0.799 | 0.159 | 0.379 | 0.675 | 0.859 | 0.914 | **0.651** | 0.744 |
| + RS | 0.817 | 0.086 | 0.201 | 0.706 | 0.795 | 0.095 | 0.342 | 0.650 | 0.856 | 0.515 | 0.615 | 0.706 |
| + DRL | **0.836** | **0.288** | **0.288** | **0.744** | **0.815** | **0.308** | **0.409** | **0.680** | **0.883** | **0.955** | 0.643 | **0.745** |
| TVAE | 0.810 | 0.325 | 0.185 | 0.717 | 0.802 | 0.351 | **0.330** | 0.686 | 0.863 | 0.858 | 0.631 | 0.750 |
| + RS | 0.788 | 0.024 | **0.237** | 0.420 | 0.778 | 0.061 | 0.283 | 0.465 | 0.846 | 0.522 | 0.480 | 0.497 |
| + DRL | **0.835** | **0.467** | 0.189 | **0.731** | **0.832** | **0.487** | **0.330** | **0.694** | **0.893** | **0.926** | **0.635** | **0.752** |
| GOGGLE | 0.622 | 0.039 | 0.248 | 0.596 | 0.648 | 0.076 | 0.296 | 0.566 | 0.742 | 0.549 | 0.551 | 0.600 |
| + RS | 0.608 | 0.047 | 0.235 | 0.577 | 0.639 | 0.084 | **0.322** | 0.549 | 0.727 | 0.571 | 0.532 | 0.592 |
| + DRL | **0.720** | **0.253** | **0.298** | **0.698** | **0.673** | **0.281** | 0.310 | **0.636** | **0.747** | **0.758** | **0.563** | **0.691** |

making the standard procedure of rejecting non-aligned samples unfeasible. Further, to visualise the impact of DRL, we consider the features *Price* and *Zipcode* from the House dataset and create the scatter plots of (i) the real data, and the synthetic data from (ii) the unconstrained DGMs, (iii) the DGMs+LL, and (iv) the DGMs+DRL. We also highlight in red the regions that violate the constraints: (i) *if the Zipcode is 98004 or 98005 then the Price is greater than 400K USD* and (ii) *if the Zipcode is between 98006 and 98008 then the Price exceeds 225K USD*. The scatter plots obtained for the real datapoints and TVAE (with and without LL and DRL) are shown in Figure 4, while the ones obtained from the other models are given in Figure 6, Appendix M. The Figures clearly show that standard DGMs and DGMs+LL fail to comply with the constraints, and indeed, many of the samples fall in the red-shaded regions. On the contrary, the samples obtained using DRL not only never violate the constraints, but also better match the real data distribution.

**Machine Learning Efficacy.** Table 2 shows that: (i) making the samples compliant with the constraints via rejection sampling (RS) often reduces their machine learning efficacy (indicated as DGMs+RS), and that (ii) adding DRL improves the performance of the unconstrained models according to at least one metric in all cases but one (TableGAN over the CCS dataset). Regarding the performance obtained with rejection sampling we can see that it decreases with respect to the standard DGMs in 17, 17 and 17 out of 20 cases for F1, $w$F1 and AUC, respectively. Regarding the performance of DGMs+DRL, the layer improves the performance w.r.t. the unconstrained models in 19, 18 and 17 out of 20 cases for F1, $w$F1 and AUC, respectively. Additionally, the improvements are often non-negligible. For F1, in more than half of the cases, the improvement is of at least 3.5%, with the largest one recorded on GOGGLE for CCS of 21.4%. For $w$F1, in more than half of the cases, the improvement is of at least 1.0%, with the largest improvement, of 20.5%, again recorded on GOGGLE for CCS. And, for AUC, the improvement is at least 1.2% in half of the cases, with the largest improvement, of 20.9%, recorded on GOGGLE for CCS. On the regression dataset, House, we find a similar trend in terms of improvements brought by DRL (Appendix K,Table 14), where the DGM+DRL models improve the performance w.r.t. the unconstrained models in all cases. We also verify the statistical significance of the results following the recommendation of (Demsar, 2006). We perform the Wilcoxon signed-rank test on the efficacy results for the classification datasets and we obtain p-value $< 0.01$ w.r.t. the F1 and $w$F1 results and $< 0.05$ w.r.t. AUC, thus confirming that DRL significantly improves the performances of DGMs.

## 4.2 LINEAR VS. QFLRA CONSTRAINTS

**Background knowledge alignment.** Table 4 shows the CVR for each DGM+LL model (first five rows) and for the DGM+DRL models (last row). As expected, DGMs+LL cannot guarantee the

Table 3: Efficacy comparison between the DGM+LL models and the models with DRL. The performance is measured using F1, $w$F1, and AUC, for each classification dataset.

| | F1 | | | | wF1 | | | | AUC | | | |
|---|---|---|---|---|---|---|---|---|---|---|---|---|
| | URL | CCS | LCLD | Heloc | URL | CCS | LCLD | Heloc | URL | CCS | LCLD | Heloc |
| WGAN+LL | **0.803** | **0.359** | 0.183 | 0.694 | 0.799 | **0.383** | 0.330 | **0.662** | 0.869 | 0.857 | 0.608 | **0.732** |
| WGAN+DRL | 0.800 | 0.313 | **0.197** | **0.721** | **0.801** | 0.340 | **0.339** | 0.652 | **0.875** | **0.885** | **0.623** | 0.717 |
| TableGAN+LL | 0.612 | **0.169** | 0.232 | **0.638** | **0.695** | **0.203** | 0.373 | **0.633** | **0.868** | **0.794** | 0.640 | 0.704 |
| TableGAN+DRL | **0.619** | 0.163 | **0.269** | 0.628 | 0.693 | 0.196 | **0.401** | 0.628 | 0.865 | 0.742 | **0.657** | **0.709** |
| CTGAN+LL | **0.836** | 0.250 | 0.265 | 0.729 | **0.820** | 0.271 | 0.392 | **0.688** | 0.880 | **0.959** | 0.641 | **0.755** |
| CTGAN+DRL | **0.836** | **0.288** | **0.288** | **0.744** | 0.815 | **0.308** | **0.409** | 0.680 | **0.883** | 0.955 | **0.643** | 0.745 |
| TVAE+LL | 0.824 | 0.413 | 0.158 | 0.730 | 0.816 | 0.436 | 0.310 | 0.691 | 0.878 | **0.933** | 0.633 | 0.747 |
| TVAE+DRL | **0.835** | **0.467** | **0.189** | **0.731** | **0.832** | **0.487** | **0.330** | **0.694** | **0.893** | 0.926 | **0.635** | **0.752** |
| GOGGLE+LL | **0.787** | 0.233 | 0.284 | **0.723** | **0.749** | 0.262 | **0.310** | **0.663** | **0.802** | **0.765** | 0.554 | **0.719** |
| GOGGLE+DRL | 0.720 | **0.253** | **0.298** | 0.698 | 0.673 | **0.281** | **0.310** | 0.636 | 0.747 | 0.758 | **0.563** | 0.691 |

compliance with QFLRA constraints and in 5 out of 25 cases we see a CVR greater than 50%. Moreover, we have one case where CVR is 100%, thus demonstrating the need for models that support more expressive constraints. In Appendix L, Tables 15, 16, 17, we report the results for all metrics: CVR, sCVC, and CVC.

**Machine Learning Efficacy.** For the sake of completeness, in Table 3 we include the comparison between DGMs+DRL and DGMs+LL on the classification datasets. Since Stoian et al. (2024) already reported improvements over their unconstrained counterpart by adding the linear layer, as expected in this scenario, we get more modest improvements than the ones w.r.t. the

Table 4: CVR for each DGM+LL model and dataset. Cases with CVR$\geq$50% are underlined. Best results are in bold.

| | URL | CCS | LCLD | Heloc | House |
|---|---|---|---|---|---|
| WGAN+LL | $8.9_{\pm 3.2}$ | $\underline{51.5_{\pm 11.2}}$ | $27.0_{\pm 3.6}$ | $20.6_{\pm 6.3}$ | $\underline{100.0_{\pm 0.0}}$ |
| TableGAN+LL | $3.6_{\pm 0.8}$ | $\underline{54.0_{\pm 17.8}}$ | $11.3_{\pm 0.9}$ | $26.6_{\pm 7.7}$ | $23.9_{\pm 2.7}$ |
| CTGAN+LL | $7.0_{\pm 2.6}$ | $\underline{55.7_{\pm 16.3}}$ | $2.6_{\pm 1.1}$ | $2.6_{\pm 2.4}$ | $10.8_{\pm 7.8}$ |
| TVAE+LL | $6.8_{\pm 0.6}$ | $8.4_{\pm 2.0}$ | $5.8_{\pm 0.8}$ | $0.0_{\pm 0.0}$ | $13.0_{\pm 12.6}$ |
| GOGGLE+LL | $6.5_{\pm 7.0}$ | $23.0_{\pm 10.7}$ | $\underline{81.9_{\pm 6.5}}$ | $11.5_{\pm 7.1}$ | $2.6_{\pm 2.6}$ |
| All + DRL | $\mathbf{0.0}_{\pm 0.0}$ | $\mathbf{0.0}_{\pm 0.0}$ | $\mathbf{0.0}_{\pm 0.0}$ | $\mathbf{0.0}_{\pm 0.0}$ | $\mathbf{0.0}_{\pm 0.0}$ |

unconstrained models. As we can see from the Table, the DGM+DRL models improve the efficacy w.r.t. the DGMs+LL for at least one metric in 17 out of 20 cases. Similarly, the number of times DGM+DRL outperforms the respective DGM+LL is lower than the number of times it outperforms its unconstrained counterpart. Indeed, out of 20 comparisons, the models with DRL outperform their linearly constrained counterparts 13, 10 and 11 times for F1, $w$F1, and AUC, respectively. Regarding the regression dataset, House, Table 21 in Appendix L shows that the DGM+DRL models have a comparable performance to the DGM+LL models, with 6 out of 10 the cases showing an improvement in performance when using our layer. As in the previous experiment, we use the Wilcoxon signed-rank test to assess whether adding DRL significantly improves over the linear layer. In this case, we obtain p-value $< 0.05$ for F1, while as expected, the test confirms that the performances of DGMs+DRL and DGMs+LL are not statistically different w.r.t. $w$F1 and AUC.

## 4.3 Sample Generation Time

To assess the impact of constraints on sample generation time, we compare the runtimes of unconstrained DGMs, DGMs+DRL and DGMs+RS. We generate 1,000 samples for each model and dataset using five different seeds and report the average runtime in Table 5 (for a detailed breakdown, see Appendix N, Table 22). As expected, DGMs+DRL are

Table 5: Sample generation time in seconds.

| | URL | CCS | LCLD | Heloc | House |
|---|---|---|---|---|---|
| DGM | 0.15 | 0.08 | 0.07 | 0.06 | 0.05 |
| DGM+RS | 0.37 | 0.83 | 1.54 | 0.66 | - |
| DGM+DRL | 0.22 | 0.13 | 0.14 | 0.10 | 0.13 |

slower on average than their unconstrained counterparts. However, they are faster than DGMs+RS. Indeed, excluding extreme cases with 100% CVR (where we were unable to generate samples even in 24h), in all other cases, DGMs+RS take more than twice as long as the unconstrained DGMs.

## 5 RELATED WORK

Our work lies at the intersection of two fields: Neuro-symbolic AI and tabular data generation. Thus, our related work section will mirror this duality.

**Neuro-symbolic AI.** Neuro-symbolic AI (Raedt et al., 2020; d'Avila Garcez & Lamb, 2023) refers to the broad area of AI that combines the strengths of symbolic reasoning with neural networks. As our work falls into the more specific field of injection of background knowledge into neural models, (see, e.g., (Stewart & Ermon, 2017; Hoernle et al., 2022; Giunchiglia et al., 2024a; Daniele et al., 2023; Calanzone et al., 2024)) we will focus the discussion on this topic. Many methods for this task are based on the intuition that logical constraints can be transformed into differentiable loss function terms that penalise the networks for violating them (see, e.g., (Xu et al., 2018; Badreddine et al., 2022; Diligenti et al., 2012; Fischer et al., 2019)). As expected, since these methods operate at a loss level, they give no guarantee that the constraints will be satisfied. Other works manage to integrate neural networks and probabilistic reasoning through the mapping of predicates appearing in logical formulae to neural networks (Manhaeve et al., 2018; Yang et al., 2020; Sachan et al., 2018; Pryor et al., 2023; van Krieken et al., 2023). This allows these methods to both perform reasoning on the networks' predictions as well as constrain the output according to the background knowledge. The most similar line of work to ours is the one where the constraints in input are automatically compiled into neural layers (Giunchiglia & Lukasiewicz, 2021; Ahmed et al., 2022; Giunchiglia et al., 2024b). However, these methods can compile and incorporate constraints that are at best as expressive as propositional logic formulae. Focusing specifically on the incorporation of constraints on generic generative models, we can find the work by Stoian et al. (2024), where the tabular generation process was simply constrained by linear inequalities. If we consider different application domains, we can find the work proposed by Misino et al. (2022), where ProbLog (Raedt et al., 2007) works in tandem with variational-autoencoders, and the one by Liello et al. (2020), where the authors incorporate propositional logic constraints on GANs for structured objects generation.

**Tabular Data Generation.** In recent years, various DGMs have been proposed to tackle the problem of tabular data synthesis. Many of these approaches are based on Generative Adversarial Networks (GANs), like TableGAN (Park et al., 2018), CTGAN (Xu et al., 2019), IT-GAN (Lee et al., 2021), OCT-GAN (Kim et al., 2021), and PacGAN (Lin et al., 2018). Other methods try to reduce the problems that often characterise GANs, such as mode collapse and unstable training, by introducing Variational AutoEncoders (VAEs) based models, see, e.g., (Xu et al., 2018; Srivastava et al., 2017; Wan et al., 2017). An alternative solution to such problems is given by the usage of denoising diffusion probabilistic models as done in (Kotelnikov et al., 2023) or (Kim et al., 2023), where the authors designed a self-paced learning method and a fine-tuning approach to adapt the standard score-based generative modeling to the challenges of tabular data generation. Finally, GOGGLE (Liu et al., 2022) uses graph learning to infer relational structure from the data and use it to their advantage especially in data-scarce setting. Since synthetic tabular data are often used to replace the original dataset to preserve privacy in sensitive settings, a parallel line of research revolves around the development of DGMs with privacy guarantees. Examples of models that have such privacy guarantees are PATEGAN (Yoon et al., 2020) and DP-CGAN (Torkzadehmahani et al., 2020).

## 6 CONCLUSIONS

In this paper, we have proposed Disjunctive Refinement Layer (DRL) the first-ever Neuro-symbolic AI layer able to automatically compile constraints expressed as QFLRA formulas into a neural layer and thus guarantee their satisfaction. This sort of work is really needed in the tabular data synthesis field, as our experimental analysis shows that Deep Generative Models (DGMs) very frequently generate datapoints that are not aligned with the background knowledge, with some extreme cases where all the datapoints are violating the constraints. DRL presents many desirable properties: (i) it can be seamlessly integrated into the topology of any neural network, (ii) it allows the backpropagation of the gradients, (iii) it performs all the computations in a single forward pass (i.e., there are no cycles), (iv) it optimally refines the original predictions and, last but not least, (v) it improves the performance of all the tested DGMs in terms of machine learning efficacy. Indeed, in our experimental analysis we got improvements for all datasets of up to 21.4%, 20.5%, and 20.9% in terms of F1-score, weighted F1-score, and Area Under the ROC Curve, respectively.

## 7 ETHICS STATEMENT

The development and application of synthetic data generation techniques, particularly in tabular data, have the potential to significantly impact a wide range of sectors, including healthcare, finance, and social sciences. While our method, Disjunctive Refinement Layer (DRL), improves the quality and fidelity of generated data by ensuring alignment with user-specified constraints, there are ethical implications of synthetic data use. Firstly, there is the potential for misuse. Synthetic data may be seen as a substitute for real-world data, but it should not be viewed as a perfect replacement. Secondly, the use of synthetic data in automated decision-making systems poses risks for fairness and bias. While DRL allows for the specification of constraints that align with real-world domain knowledge, it is important that the user-specified constraints do not encode existing biases or discrimination.

## 8 REPRODUCIBILITY STATEMENT

To ensure the reproducibility of our results, we included all the necessary details in the Appendix of the paper. The proofs of the Lemmas and Theorems can be found in Appendices B, C, D, E, and F, the detailed description of the datasets, the baseline models used (together with their links), evaluation protocol for the machine learning efficacy metric, and the chosen hyperparameters can be found in G, H, I, and J.

### ACKNOWLEDGMENTS

Mihaela Cătălina Stoian is supported by the EPSRC under the grant EP/T517811/1. She has also received support for this work through the G-Research Women in Quant Finance Grant and St Hilda's College Travel for Research and Study Grant. We also acknowledge the use of the Advanced Research Computing (ARC) facilities of University of Oxford.

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

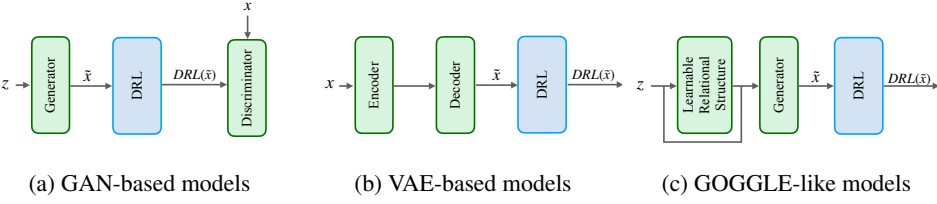

(a) GAN-based models    (b) VAE-based models    (c) GOGGLE-like models

Figure 5: Visualisation of the considered types of DGMs and how to add DRL in their topology.

## A  Disjunctive Refinement Layer Visualizations

In Figure 5 we give an overview on how to add DRL in the topology of the three types of models we considered. In all figures we indicate with $z$ a noise vector, with $x$ a real datapoint from the original dataset, with $\tilde{x}$ a sample generated with the DGM, and with $DRL(\tilde{x})$ the final sample obtained from DRL. Considering each of the Figures, we can see that:

- Figure 5a shows that DRL needs to be added on top of the generator module in GAN-based models,
- Figure 5b shows that DRL needs to be added after the decoder module in VAE-based models, and
- Figure 5c shows that DRL needs to be added after the generator module in GOGGLE-like models.

In general, we can see that DRL can be added in many different DGMs, and it simply needs to be added right after the sample $\tilde{x}$ is generated.

## B  Proof of Lemma 3.1

**Lemma.** *Let $\Pi$ be a finite and satisfiable set of constraints in a single variable $x_i$. For every sample $\tilde{x}$, $DRL(\tilde{x})$ satisfies $\Pi$ and is optimal wrt $\tilde{x}$.*

*Proof.* We first prove that for every sample $\tilde{x}$, $DRL(\tilde{x})_i$ always satisfies $\Pi$, and then that for every sample $\tilde{x}$, $DRL(\tilde{x})_i$ is the solution of $\Pi$ with minimal Euclidean distance from $\tilde{x}$.

Suppose there exists a sample $\tilde{x}$ such that $DRL(\tilde{x}) \notin \Omega(\Pi)$. This entails (i) that $\Pi \neq \emptyset$ and (ii) that $\tilde{x} \notin \Omega(\Pi)$. Since $\Pi \neq \emptyset$ and $\Pi$ is satisfiable, $l_i^{\Pi}(\tilde{x}) \neq -\infty$ or $r_i^{\Pi}(\tilde{x}) \neq +\infty$, and $DRL(\tilde{x})_i = l_i^{\Pi}(\tilde{x})$ or $DRL(\tilde{x})_i = r_i^{\Pi}(\tilde{x})$. Since by definition $l_i^{\Pi}(\tilde{x})$ and $r_i^{\Pi}(\tilde{x})$ satisfy $\Pi$ we reached a contradiction.

Assume $\tilde{x} \notin \Omega(\Pi)$ (otherwise we would have again $DRL(\tilde{x}) = \tilde{x}$ and the thesis would trivially hold). Let $d$ be the minimum Euclidean distance between any point in $\Omega(\Pi)$ and $\tilde{x}$. Let $r$ and $l$ be the two samples with $r_k = l_k = \tilde{x}_k = DRL(\tilde{x})_k$ when $k \neq i$ and $k \in \{1, \ldots, D\}$, $r_i = \tilde{x}_i + d$ and $l_i = \tilde{x}_i - d$. Either $r$ or $l$ or both belong to $\Omega(\Pi)$. Let $v$ be $r$ if $r \in \Omega(\Pi)$, and $l$ otherwise. By definition, $v \in \Omega(\Pi)$ and is optimal wrt $\tilde{x}$. Assume $v = l$. Then, from the optimality of $v$, we have that for every $v'$ with $v'_i \in (v_i, \tilde{x}_i + d)$, $v' \notin \Omega(\Pi)$. Hence, there must exist a constraint $\Psi$ such that $v_i = l_i^{\Psi}$ and thus $v_i = DRL(\tilde{x})_i$. Analogously for the case $v = r$.  $\square$

## C  Proof of Lemma 3.2

**Lemma.** *The CP resolution rule is sound: the premises entail the conclusion of the rule.*

*Proof.* Consider the CP resolution rule (7), reported below for simplicity:

$$\frac{\bigvee_{j=1}^{m}(w'_j x_i + \varphi'_j \geq 0) \vee \Phi' \qquad \bigvee_{k=1}^{n}(w_k x_i + \varphi_k \geq 0) \vee \Phi}{\bigvee_{j=1}^{m}\bigvee_{k=1}^{n}(\varphi_k/w_k - \varphi'_j/w'_j \geq 0) \vee \Phi \vee \Phi'}.$$

with $w'_1, \ldots, w'_m < 0 < w_1, \ldots, w_n$ and $m, n \geq 1$. We have to show that any model $\tilde{x}$ of the premises is also a model of the conclusion. Assuming $\tilde{x}$ satisfies the premises and not $(\Phi \vee \Phi')$ (otherwise the thesis trivially holds), it must be the case that:

$$\tilde{x}_i \geq \min_{k=1}^{n} -\tilde{x}(\varphi_k/w_k) \qquad \text{and} \qquad \tilde{x}_i \leq \max_{j=1}^{m} -\tilde{x}(\varphi'_j/w_j),$$

where, given a linear expression $\varphi$, $\tilde{x}(\varphi)$ is the application of $\tilde{x}$ to $\varphi$, i.e., the value obtained by replacing each variable $x_j$ with $\tilde{x}_j$ in $\varphi$. The above is possible if and only if

$$\min_{k=1}^{n} -\tilde{x}(\varphi_k/w_k) \leq \max_{j=1}^{m} -\tilde{x}(\varphi'_j/w_j),$$

i.e., there exist a pair $(j, k)$ such that $(-\tilde{x}(\varphi_k/w_k) \leq -\tilde{x}(\varphi'_j/w_j))$, and hence the thesis. $\qquad \square$

## D    PROOF OF LEMMA 3.3

**Lemma.** *Let $\Pi$ be a set of constraints in the variables $x_1, \ldots, x_i$. $\Pi_i = \Pi$ and $\Pi_{i-1}$ are equisatisfiable, and each assignment to the variables $x_1, \ldots, x_{i-1}$ satisfying $\Pi_{i-1}$ can be extended in order to satisfy $\Pi_i$.*

*Proof.* Clearly, given the soundness of the CP resolution rule, if $\Pi_i$ is satisfiable, then also $\Pi_{i-1}$ is satisfiable (each constraint in $\Pi_{i-1}$ and not in $\Pi_i$ is entailed by $\Pi_i$).

It remains to show that if $\tilde{x}^{:i}$ is an assignment to the variables $x_1, \ldots, x_{i-1}$ satisfying $\Pi_{i-1}$, the set of constraints $\tilde{x}^{:i}(\Pi_i)$ is satisfiable. Similarly to the notation used in the proof of lemma 3.2 in Appendix C, given a set of constraints $\Pi$, the expression $\tilde{x}^{:i}(\Pi)$ denotes the set of constraints in the variable $x_i$ obtained by substituting each variable $x_j$ ($j < i$) with the corresponding value $\tilde{x}^{:i}_j$ in the constraints in $\Pi$.

Assume $\tilde{x}^{:i}(\Pi_i)$ is not satisfiable. Then, there exist two constraints $\Psi$ and $\Psi'$ in $\tilde{x}^{:i}(\Pi_i)$ equivalent to $(x_i \geq r_i)$ and $(x_i \leq l_i)$, respectively, and

1. either $l_i < r_i$,

2. or $l_i \geq r_i$ and there exists $n \geq 1$ constraints $\{\Psi_1, \Psi_2, \ldots, \Psi_n\}$ in $\tilde{x}^{:i}(\Pi_i)$ with each $\Psi_j$ equivalent to $(x_i \leq l_i^{\Psi_j}) \vee (x_i \geq r_i^{\Psi_j})$ and $l_i^{\Psi_1}, l_i^{\Psi_2}, \ldots, l_i^{\Psi_n}, r_i^{\Psi_1}, r_i^{\Psi_2}, \ldots, r_i^{\Psi_n}$ such that $l_i^{\Psi_1} < r_i \leq r_i^{\Psi_1}$, $l_i^{\Psi_2} < r_i^{\Psi_1} \leq r_i^{\Psi_2}$, $\ldots$, $l_i^{\Psi_n} < r_i^{\Psi_{n-1}} \leq l_i < r_i^{\Psi_n}$ and thus $l_i^{\Psi_1} < r_i \leq l_i < r_i^{\Psi_n}$.

However, $l_i < r_i$ is not possible because $CPres_i(\Psi, \Psi')$ belongs to $\tilde{x}^{:i}(\Pi_{i-1})$ and is equivalent to $(r_i \leq l_i)$. Regarding the second case, $\tilde{x}^i(\Pi_i^{\ddagger})$ contains the constraints ($\equiv$ denotes logical equivalence)

$$\Upsilon_1 = CPres_i(\Psi, \Psi_1) \equiv (x_i \geq r_i^{\Psi_1}) \vee (r_i \leq l_i^{\Psi_1}) \equiv x_i \geq r_i^{\Psi_1},$$
$$\Upsilon_2 = CPres_i(\Upsilon_1, \Psi_2) \equiv (x_i \geq r_i^{\Psi_2}) \vee (r_i^{\Psi_1} \leq l_i^{\Psi_2}) \equiv x_i \geq r_i^{\Psi_2},$$
$$\ldots,$$
$$\Upsilon_n = CPres_i(\Upsilon_{n-1}, \Psi_n) \equiv x_i \geq r_i^{\Psi_n},$$

and thus $\tilde{x}^{:i}(\Pi_{i-1})$ contains $CPres_i(\Upsilon_n, \Psi') \equiv r_i^{\Psi_n} \leq l_i$, thus reaching a contradiction. $\qquad \square$

## E    PROOF OF THEOREM 3.5

**Theorem.** *Let $\Pi$ be a finite and satisfiable set of constraints. For any sample $\tilde{x}$ and variable ordering, the corresponding sample $DRL(\tilde{x})$ satisfies $\Pi$.*

*Proof.* We prove the statement by induction over the number $n$ of variables appearing in $\Pi$.

Let $n = 0$. In this case $\Pi$ is satisfied by any sample $\tilde{x}$, and $DRL(\tilde{x}) = \tilde{x}$.

Let $n > 1$. Let $x_i$ be the last variable in the ordering occurring in $\Pi$. Since $\Pi_{i-1}$ contains $(n-1)$ variables, DRL$(\tilde{x})$ satisfies $\Pi_{i-1}$ by the inductive hypothesis. From Lemma 3.3 we know that $\widetilde{\Pi}_i$ is satisfiable, and hence the thesis follows from Lemma 3.1. □

## F  PROOF OF THEOREM 3.6

**Theorem.** *Let $\Pi$ be a finite and satisfiable set of constraints. For any sample $\tilde{x}$ and variable ordering, the corresponding sample DRL$(\tilde{x})$ is optimal wrt $\tilde{x}$, $\Pi$ and the variable ordering.*

*Proof.* We prove the statement by induction over the number $n$ of variables occurring in $\Pi$.

Let $n = 0$. In this case $\Pi$ is satisfied by any sample $\tilde{x}$, and DRL$(\tilde{x}) = \tilde{x}$.

Let $n > 1$. Let $x_i$ be the last variable in the ordering occurring in $\Pi$. Since $\Pi_{i-1}$ contains only the variables $x_1, x_2, \ldots x_{i-1}$, we know that for any sample $\tilde{x}$, DRL$(\tilde{x})$ is optimal with respect to $\tilde{x}$, $\Pi_{i-1}$ and the variable ordering for the inductive hypothesis. From Lemma 3.3 we know that $\widetilde{\Pi}_i$ is satisfiable. From Lemma 3.1 we know that for every $\tilde{x}$, DRL$(\tilde{x})$ is optimal wrt to $\tilde{x}$ and $\widetilde{\Pi}_i$, and hence the thesis. □

## G  DATASETS

Below we provide a brief description for each dataset and the links to the pages where they can be downloaded.

- URL[3] (Hannousse & Yahiouche, 2021) is used to perform webpage phishing detection with features describing statistical properties of the URL itself as well as the content of the page.
- CCS[4] is used to identify individuals at high risk of cervical cancer from features describing the patients' demographic and medical history, including age, sexual behavior, contraceptive use, and various medical test results.
- LCLD[5] is used to predict whether the debt lent is unlikely to be collected from features related to the loan as well as client history. In particular, we use the feature-engineered dataset from Simonetto et al. (2022), inspired from the LendingClub loan data.
- HELOC[6] is a dataset from FICO used to predict whether customers will repay their credit lines within 2 years from features related to the credit line and the client's history.
- House[7] was used to predict the prices of houses in King County (USA) and contains data collected from May 2014 to May 2015. The features describe various features of the sold houses, including the date of sale, house prices, the number of bedrooms and bathrooms, square footage, condition, grade, year built, and location, among others.

For each dataset above, Table 6 shows the number of samples in the train, validation and test partitions, along with the number of features and the number of constraints. Regarding the constraints, they were already included in some of the original datasets. This is true for URL, Heloc and LCLD. Regarding CCS and House, we manually annotated the constraints using our background knowledge about the problem, then we checked whether the data were compliant with our constraints and finally we retained only those constraints that were satisfied by all the datapoints. Simple examples of these constraints (from CCS) state simple facts like "if the feature capturing the number of cigarettes packs per day is greater than 0 then the feature capturing whether the patient smokes or not should be equal to 1" or "the feature representing the age should always be higher than the age at the first intercourse".

---

[3]Link to dataset: https://data.mendeley.com/datasets/c2gw7fy2j4/2
[4]Link to dataset:https://www.kaggle.com/datasets/ranzeet013/cervical-cancer-dataset/data
[5]Link to dataset: https://figshare.com/s/84ae808ce6999fafd192
[6]Link to dataset: https://huggingface.co/datasets/mstz/heloc
[7]Link to dataset: https://www.kaggle.com/datasets/harlfoxem/housesalesprediction/data

Table 6: Dataset statistics.

| Dataset | # Train | # Val | # Test | # Features | # Constraints | Task (# classes) |
|---------|---------|-------|--------|-----------|---------------|-------------------|
| URL | 7K | 2K | 2K | 64 | 18 | Binary classification |
| CCS | 1K | 0.02K | 0.15K | 36 | 12 | Binary classification |
| LCLD | 494K | 199K | 431K | 29 | 16 | Binary classification |
| Heloc | 8K | 2K | 0.2K | 24 | 12 | Binary classification |
| House | 17K | 0.5K | 4K | 20 | 13 | Regression |

## H   MODELS

Below we give a brief description of each of the models used in our experimental analysis:

- WGAN (Arjovsky et al., 2017): it is a GAN based model which has been trained by using the Wasserstein distance as a loss, which improves the stability of learning.

- TableGAN (Park et al., 2018): is a GAN-based model designed for generating realistic tabular data. It uses a convolutional neural network (CNN) as a discriminator to better the capture dependencies among features.

- CTGAN (Xu et al., 2019): is again a GAN-based model which uses a conditional generator to model feature distributions and applies a mode-specific normalisation technique to improve the generation of imbalanced categorical data.

- TVAE (Xu et al., 2019): uses a variational autoencoder architecture to capture both continuous and categorical feature distributions, learning a probabilistic latent space representation of the data. By optimising the evidence lower bound, it balances reconstruction accuracy and regularisation.

- GOGGLE (Liu et al., 2022): uses a VAE framework combined with a graph neural network, which allows the model to learn complex feature relationships by representing the data as a graph, where each node corresponds to a feature, and edges capture dependencies between features.

## I   EFFICACY EVALUATION PROTOCOL

In order to evaluate the efficacy of the models, we closely follow the protocol outlined in (Kim et al., 2023). For clarity, we describe the protocol below.

First, we generate a synthetic dataset and split it into training, validation, and test sets, maintaining the same proportions as in the real dataset. Next, we conduct a hyperparameter search using the synthetic training set to train various classifiers and regressors. Specifically, for the classification datasets (i.e., URL, CCS, LCLD, Heloc), we use the following classifiers: AdaBoost (Schapire, 2013), Decision Tree (Wu et al., 2008), Logistic Regression (Cox, 1958) Multi-layer Perceptron (MLP) (Haykin, 1994), Random Forest (Ho, 1995), and XGBoost (Chen & Guestrin, 2016). For the regression dataset (i.e., House), we use: Linear Regression, MLP, Random Forest regressors, and XGBoost. Across all classifiers and regressors, we use the same hyperparameter settings as those in Table 26 of (Kim et al., 2023). Then, based on the F1-score obtained on the real validation set, we select the best hyperparameter configuration. As a last step, we evaluate the selected models on the real test set and average the performance across all classifiers/regressors. The results for all models are reported using three metrics (i.e., F1-score, weighted F1-score, and the Area Under the ROC Curve) for the classification datasets, and two metrics (i.e., Mean Absolute Error and Root Mean Square Error) for the regression dataset.

We run the entire process five times for each model, then compute the average results for each metric individually across the repetitions.

Table 7: Best hyperparameter settings used for DGMs (and also for DGMs+LL and DGMs+DRL) in our experiments.

| Model/Dataset | Hyperparameter | URL | CCS | LCLD | Heloc | House |
|---|---|---|---|---|---|---|
| WGAN | Batch size | 510 | 256 | 510 | 510 | 510 |
| | Optimiser | Adam | Adam | Adam | Adam | Adam |
| | Learning rate | 0.001 | 0.001 | 0.001 | 0.001 | 0.0002 |
| | Epochs | 150 | 1250 | 15 | 150 | 100 |
| | Discriminator iters | 5 | 5 | 5 | 5 | 1 |
| | LL Ordering | Corr | KDE | Rnd | Corr | Rnd |
| | DRL Ordering | Rnd | Rnd | KDE | Rnd | KDE |
| TableGAN | Batch size | 128 | 128 | 510 | 128 | 256 |
| | Optimiser | Adam | Adam | Adam | Adam | Adam |
| | Learning rate | 0.001 | 0.0002 | 0.01 | 0.001 | 0.0001 |
| | Epochs | 300 | 2000 | 20 | 200 | 50 |
| | LL Ordering | Corr | KDE | KDE | Corr | KDE |
| | DRL Ordering | KDE | KDE | KDE | Corr | Rnd |
| CTGAN | Batch size | 500 | 70 | 500 | 500 | 500 |
| | Optimiser | Adam | Adam | Adam | Adam | Adam |
| | Learning rate | 0.0002 | 0.001 | 0.0002 | 0.0002 | 0.0002 |
| | Epochs | 150 | 1000 | 20 | 500 | 150 |
| | LL Ordering | KDE | KDE | KDE | Corr | Rnd |
| | DRL Ordering | KDE | Corr | Corr | Corr | Rnd |
| TVAE | Batch size | 70 | 70 | 500 | 500 | 70 |
| | Optimiser | Adam | Adam | Adam | Adam | Adam |
| | Learning rate | 0.0002 | 0.0001 | 0.00001 | 0.000005 | 0.0002 |
| | Epochs | 150 | 1500 | 40 | 150 | 150 |
| | Loss factor | 2 | 2 | 4 | 2 | 2 |
| | LL Ordering | KDE | Rnd | Corr | KDE | Rnd |
| | DRL Ordering | Rnd | Rnd | Corr | Corr | KDE |
| GOGGLE | Batch size | 128 | 32 | 128 | 64 | 64 |
| | Optimiser | Adam | Adam | Adam | Adam | Adam |
| | Learning rate | 0.005 | 0.01 | 0.001 | 0.001 | 0.01 |
| | Epochs | 1000 | 500 | 60 | 1000 | 400 |
| | Threshold | 0.1 | 0.2 | 0.2 | 0.1 | 0.2 |
| | Patience | 50 | 50 | 50 | 50 | 50 |
| | LL Ordering | KDE | Rnd | Rnd | Rnd | Rnd |
| | DRL Ordering | Rnd | Rnd | Rnd | Rnd | Rnd |

## J   HYPERPARAMETER SEARCH

We carried out an extensive hyperparameter search to identify the optimal configurations for each DGM. We selected these configurations based on the efficacy performance: for the classification datasets (i.e., URL, CCS, LCLD, and Heloc), we used the average of the F1-score, weighted F1-score, and Area Under the ROC Curve (AUC), whereas for the regression dataset (i.e., House), we used the average of the Mean Absolute Error and Root Mean Square Error.

For clarity, we describe the hyperparameter search space below, for each of the considered models. For the GOGGLE model, we adopted the same optimiser and learning rate settings as in (Liu et al., 2022). Specifically, we used the Adam optimizer (Kingma & Ba, 2015) with five learning rates: $1 \times 10^{-3}, 5 \times 10^{-3}, 1 \times 10^{-2}$. Additionally, we experimented with a set of values for the threshold parameter: $\{1 \times 10^{-1}, 2 \times 10^{-1}\}$. For the TVAE model, Adam was used again, but with a different set of five learning rates: $5 \times 10^{-6}, 1 \times 10^{-5}, 1 \times 10^{-4}, 2 \times 10^{-4}, 1 \times 10^{-3}$. For the other three models (i.e., WGAN, TableGAN, and CTGAN), we tested three different optimisers: Adam, RM-SProp (Hinton, 2014), and SGD (Ruder, 2016), each paired with its own set of learning rates, as follows:

Table 8: Constraint violation rate (CVR) for each unconstrained DGM model and each dataset.

| Constraint Type | Model/Dataset | URL | CCS | LCLD | Heloc | House |
|---|---|---|---|---|---|---|
| Linear | WGAN | 17.9±5.0 | 15.0±5.6 | 28.5±16.3 | 69.1±8.6 | 100.0±0.0 |
| | TableGAN | 5.4±1.4 | 14.5±3.6 | 19.1±3.7 | 45.6±16.3 | 100.0±0.0 |
| | CTGAN | 3.8±1.3 | 56.1±7.5 | 1.9±1.1 | 55.8±10.2 | 100.0±0.0 |
| | TVAE | 3.0±0.7 | 8.6±1.9 | 3.9±0.5 | 44.8±1.0 | 100.0±0.0 |
| | GOGGLE | 47.3±6.9 | 42.5±3.9 | 16.5±13.2 | 47.3±6.9 | 99.9±0.1 |
| | All models + DRL | **0.0**±0.0 | **0.0**±0.0 | **0.0**±0.0 | **0.0**±0.0 | **0.0** ±0.0 |
| Disjunctive | WGAN | 8.6±1.7 | 38.3±10.0 | 26.8±5.2 | 47.3±15.5 | 74.2±4.4 |
| | TableGAN | 3.9±1.4 | 56.1±14.4 | 16.0±3.5 | 33.8±14.7 | 73.7±14.8 |
| | CTGAN | 6.3±1.0 | 58.8±6.4 | 5.3±0.8 | 1.7±1.4 | 42.8±23.1 |
| | TVAE | 7.6±0.7 | 11.8±0.7 | 6.6±0.5 | 0.0±0.0 | 52.7±24.7 |
| | GOGGLE | 2.0±2.8 | 37.1±15.8 | 65.3±15.8 | 35.2±4.2 | 20.4±20.5 |
| | All models + DRL | **0.0**±0.0 | **0.0**±0.0 | **0.0**±0.0 | **0.0**±0.0 | **0.0** ±0.0 |

- WGAN: Adam: $\{1\times10^{-4}, 2\times10^{-4}, 1\times10^{-3}\}$, RMSProp: $\{5\times10^{-5}, 1\times10^{-4}, 1\times10^{-3}\}$, SGD: $\{1 \times 10^{-4}, 1 \times 10^{-3}\}$

- TableGAN: Adam: $\{5 \times 10^{-5}, 1 \times 10^{-4}, 2 \times 10^{-4}, 1 \times 10^{-3}, 1 \times 10^{-2}\}$, RMSProp: $\{1 \times 10^{-4}, 2 \times 10^{-4}, 1 \times 10^{-3}\}$, SGD: $\{1 \times 10^{-4}, 1 \times 10^{-3}\}$

- CTGAN: Adam: $\{5 \times 10^{-5}, 1 \times 10^{-4}, 2 \times 10^{-4}, 1 \times 10^{-3}\}$, RMSProp: $\{1 \times 10^{-4}, 2 \times 10^{-4}, 1 \times 10^{-3}\}$, SGD: $\{1 \times 10^{-4}, 1 \times 10^{-3}\}$

Additionally, we explored various batch sizes for each model:

- WGAN: $\{64, 128, 256, 510\}$

- TableGAN: $\{128, 256, 510\}$

- CTGAN and TVAE: $\{70, 280, 500\}$

- GOGGLE: $\{32, 64, 128\}$

We did not separately tune our DGM+DRL models, nor the DGM+LL models. However, for each of the DGM+DRL and DGM+LL models, we ran three versions corresponding to three different ways of ordering the variables that decided the order in which the layers (LL and DRL) change the values of the features. More precisely, we tried: (i) a random (Rnd) ordering, (ii) the correlation (Corr)-based ordering. (iii) the Kernel Density Estimation (KDE)-based ordering. The last two orderings are proposed by Stoian et al. (2024), and are defined in the Appendix of their paper. For completeness, we also define them here. The Corr-based ordering is computed as follows: for each feature, the absolute difference is taken between the pairwise feature correlations (with respect to all other features) of samples generated by the unconstrained DGM and the real data. The features are then ranked in ascending order based on these scores. This ensures that features with the most similar correlations between the generated and real data are prioritised by DRL (and LL) first. The KDE-based ordering is computed by first fitting a Kernel Density Estimator (KDE) on the real data and estimating the log-likelihood for each real and synthetic sample. In a discrete setting, regardless of the variable domains, two marginal probability mass functions are approximated for each variable using the real and synthetic data, respectively. The variables are then ranked by computing the Kullback-Leibler divergence between these two and sorting the results in ascending order. We then selected the best ordering separately for each DGM+DRL and DGM+LL model. In Table 7, we report the optimal hyperparameter configurations, which we use in all experiments presented in our paper that involve the DGM, DGM+LL and DGM+DRL models.

## K SYNTHETIC DATA QUALITY.

**Background knowledge alignment.** In addition to the CVR metric reported in the main paper, we also compute the samplewise constraints violation coverage (sCVC) and the constraint violation coverage (CVC), where sCVC indicates the average percentage of constraints violated per sample,

Table 9: Samplewise constraints violation coverage (sCVC) for each unconstrained DGM model and each dataset.

| Constraint Type | Model/Dataset | URL | CCS | LCLD | Heloc | House |
|---|---|---|---|---|---|---|
| Linear | WGAN | $2.2_{\pm0.6}$ | $7.9_{\pm2.9}$ | $15.1_{\pm9.4}$ | $14.3_{\pm2.5}$ | $50.0_{\pm0.0}$ |
| | TableGAN | $0.7_{\pm0.2}$ | $7.5_{\pm1.9}$ | $9.8_{\pm1.9}$ | $9.0_{\pm3.3}$ | $50.0_{\pm0.0}$ |
| | CTGAN | $0.5_{\pm0.2}$ | $32.1_{\pm5.6}$ | $1.0_{\pm0.5}$ | $9.9_{\pm2.8}$ | $50.0_{\pm0.0}$ |
| | TVAE | $0.4_{\pm0.1}$ | $4.4_{\pm1.0}$ | $2.0_{\pm0.3}$ | $7.4_{\pm0.2}$ | $50.0_{\pm0.0}$ |
| | GOGGLE | $17.2_{\pm4.9}$ | $22.0_{\pm2.3}$ | $8.6_{\pm7.1}$ | $17.2_{\pm4.9}$ | $50.0_{\pm0.0}$ |
| | All models + DRL | $\mathbf{0.0}_{\pm0.0}$ | $\mathbf{0.0}_{\pm0.0}$ | $\mathbf{0.0}_{\pm0.0}$ | $\mathbf{0.0}_{\pm0.0}$ | $\mathbf{0.0}_{\pm0.0}$ |
| Disjunctive | WGAN | $0.9_{\pm0.2}$ | $5.3_{\pm1.7}$ | $2.2_{\pm0.5}$ | $11.5_{\pm4.4}$ | $7.0_{\pm0.4}$ |
| | TableGAN | $0.4_{\pm0.1}$ | $6.8_{\pm1.9}$ | $1.5_{\pm0.3}$ | $7.9_{\pm3.7}$ | $6.7_{\pm1.3}$ |
| | CTGAN | $0.6_{\pm0.1}$ | $8.0_{\pm1.2}$ | $0.4_{\pm0.1}$ | $0.3_{\pm0.3}$ | $3.9_{\pm2.1}$ |
| | TVAE | $0.8_{\pm0.1}$ | $1.3_{\pm0.1}$ | $0.5_{\pm0.0}$ | $0.0_{\pm0.0}$ | $4.8_{\pm2.3}$ |
| | GOGGLE | $0.2_{\pm0.3}$ | $5.0_{\pm2.1}$ | $10.0_{\pm3.4}$ | $7.1_{\pm0.9}$ | $1.9_{\pm1.9}$ |
| | All models + DRL | $\mathbf{0.0}_{\pm0.0}$ | $\mathbf{0.0}_{\pm0.0}$ | $\mathbf{0.0}_{\pm0.0}$ | $\mathbf{0.0}_{\pm0.0}$ | $\mathbf{0.0}_{\pm0.0}$ |

Table 10: Constraints violation coverage (CVC) for each unconstrained DGM model and each dataset.

| Constraint Type | Model/Dataset | URL | CCS | LCLD | Heloc | House |
|---|---|---|---|---|---|---|
| Linear | WGAN | $45.0_{\pm5.0}$ | $100.0_{\pm0.0}$ | $100.0_{\pm0.0}$ | $100.0_{\pm0.0}$ | $100.0_{\pm0.0}$ |
| | TableGAN | $34.0_{\pm4.2}$ | $100.0_{\pm0.0}$ | $100.0_{\pm0.0}$ | $100.0_{\pm0.0}$ | $100.0_{\pm0.0}$ |
| | CTGAN | $17.5_{\pm4.3}$ | $100.0_{\pm0.0}$ | $100.0_{\pm0.0}$ | $100.0_{\pm0.0}$ | $100.0_{\pm0.0}$ |
| | TVAE | $12.5_{\pm0.0}$ | $100.0_{\pm0.0}$ | $100.0_{\pm0.0}$ | $99.4_{\pm1.3}$ | $100.0_{\pm0.0}$ |
| | GOGGLE | $100.0_{\pm0.0}$ | $100.0_{\pm0.0}$ | $100.0_{\pm0.0}$ | $100.0_{\pm0.0}$ | $100.0_{\pm0.0}$ |
| | All models + DRL | $\mathbf{0.0}_{\pm0.0}$ | $\mathbf{0.0}_{\pm0.0}$ | $\mathbf{0.0}_{\pm0.0}$ | $\mathbf{0.0}_{\pm0.0}$ | $\mathbf{0.0}_{\pm0.0}$ |
| Disjunctive | WGAN | $50.0_{\pm0.0}$ | $40.0_{\pm0.0}$ | $28.0_{\pm1.3}$ | $80.0_{\pm0.0}$ | $34.9_{\pm2.4}$ |
| | TableGAN | $53.6_{\pm3.8}$ | $40.0_{\pm0.0}$ | $51.3_{\pm3.9}$ | $80.0_{\pm0.0}$ | $36.4_{\pm0.0}$ |
| | CTGAN | $56.4_{\pm5.0}$ | $40.0_{\pm0.0}$ | $22.3_{\pm2.2}$ | $55.2_{\pm6.6}$ | $34.9_{\pm3.3}$ |
| | TVAE | $55.2_{\pm4.6}$ | $40.0_{\pm0.0}$ | $20.3_{\pm3.4}$ | $24.8_{\pm11.1}$ | $36.0_{\pm0.8}$ |
| | GOGGLE | $28.4_{\pm21.3}$ | $40.0_{\pm0.0}$ | $49.1_{\pm8.1}$ | $48.8_{\pm15.6}$ | $33.3_{\pm5.2}$ |
| | All models + DRL | $\mathbf{0.0}_{\pm0.0}$ | $\mathbf{0.0}_{\pm0.0}$ | $\mathbf{0.0}_{\pm0.0}$ | $\mathbf{0.0}_{\pm0.0}$ | $\mathbf{0.0}_{\pm0.0}$ |

across all samples, and CVC represents the percentage of constraints violated at least once by the samples. In Tables 8, 9, and 10, we provide the CVR, sCVC, and CVC, respectively, for the unconstrained DGM models, according to the two possible types: linear and disjunctive constraints. Indeed, as constraints expressed as linear inequalities are a special case of the QFLRA constraints, we have also partitioned the constraints between those that can be captured as linear inequalities and those that cannot. Then, we checked how much each of the two partitions contributes to the high CVR results, and found that in 13 (resp. 6) out of 25 cases, the CVR was greater than 25% (resp. 50%) on the constraints presenting disjunctions alone.

**Efficacy.** Tables 11, 12, 13 show the efficacy, along with the standard deviations from the mean, for each unconstrained DGM model and the corresponding DGM+DRL models on every classification dataset, using the F1, $w$F1, and AUC metrics, respectively. Similarly, in Table 14, we report MAE and RMSE for each unconstrained DGM model and the corresponding DGM+DRL models on the regression dataset (i.e., House).

## L  LINEAR VS. QFLRA CONSTRAINTS

**Background knowledge alignment.** In Tables 15, 16, and 17, we provide the CVR, sCVC, and CVC, respectively, for the DGM+LL models, according to the two possible types: linear and disjunctive constraints.

Table 11: Efficacy comparison between the unconstrained DGM models and their DRL counterparts in terms of F1. The performance, along with the standard deviation, is reported for each classification dataset.

|  | URL | CC | LCLD | HELOC |
|---|---|---|---|---|
| WGAN | $0.794_{\pm 0.041}$ | $0.303_{\pm 0.060}$ | $0.139_{\pm 0.053}$ | $0.665_{\pm 0.050}$ |
| WGAN+RS | $0.792_{\pm 0.031}$ | $0.051_{\pm 0.037}$ | $0.156_{\pm 0.074}$ | $0.628_{\pm 0.043}$ |
| WGAN+DRL | $\mathbf{0.800}_{\pm 0.011}$ | $\mathbf{0.313}_{\pm 0.127}$ | $\mathbf{0.197}_{\pm 0.060}$ | $\mathbf{0.721}_{\pm 0.027}$ |
| TableGAN | $0.562_{\pm 0.051}$ | $\mathbf{0.196}_{\pm 0.037}$ | $0.259_{\pm 0.011}$ | $0.593_{\pm 0.058}$ |
| TableGAN+RS | $0.544_{\pm 0.071}$ | $0.138_{\pm 0.025}$ | $0.251_{\pm 0.020}$ | $0.568_{\pm 0.077}$ |
| TableGAN+DRL | $\mathbf{0.619}_{\pm 0.046}$ | $0.163_{\pm 0.079}$ | $\mathbf{0.269}_{\pm 0.025}$ | $\mathbf{0.628}_{\pm 0.083}$ |
| CTGAN | $0.822_{\pm 0.017}$ | $0.145_{\pm 0.040}$ | $0.247_{\pm 0.087}$ | $0.736_{\pm 0.035}$ |
| CTGAN+RS | $0.817_{\pm 0.008}$ | $0.086_{\pm 0.016}$ | $0.201_{\pm 0.066}$ | $0.706_{\pm 0.014}$ |
| CTGAN+DRL | $\mathbf{0.836}_{\pm 0.004}$ | $\mathbf{0.288}_{\pm 0.116}$ | $\mathbf{0.288}_{\pm 0.013}$ | $\mathbf{0.744}_{\pm 0.020}$ |
| TVAE | $0.810_{\pm 0.008}$ | $0.325_{\pm 0.190}$ | $0.185_{\pm 0.021}$ | $0.717_{\pm 0.013}$ |
| TVAE+RS | $0.788_{\pm 0.023}$ | $0.024_{\pm 0.011}$ | $\mathbf{0.237}_{\pm 0.018}$ | $0.420_{\pm 0.007}$ |
| TVAE+DRL | $\mathbf{0.835}_{\pm 0.009}$ | $\mathbf{0.467}_{\pm 0.100}$ | $0.189_{\pm 0.022}$ | $\mathbf{0.731}_{\pm 0.009}$ |
| GOGGLE | $0.622_{\pm 0.094}$ | $0.039_{\pm 0.016}$ | $0.248_{\pm 0.156}$ | $0.596_{\pm 0.072}$ |
| GOGGLE+RS | $0.608_{\pm 0.098}$ | $0.047_{\pm 0.024}$ | $0.235_{\pm 0.149}$ | $0.577_{\pm 0.093}$ |
| GOGGLE+DRL | $\mathbf{0.720}_{\pm 0.086}$ | $\mathbf{0.253}_{\pm 0.144}$ | $\mathbf{0.298}_{\pm 0.153}$ | $\mathbf{0.698}_{\pm 0.023}$ |

Table 12: Efficacy comparison between the unconstrained DGM models and their DRL counterparts in terms of $w$F1. The performance, along with the standard deviation, is reported for each classification dataset.

|  | URL | CC | LCLD | HELOC |
|---|---|---|---|---|
| WGAN | $0.796_{\pm 0.026}$ | $0.330_{\pm 0.057}$ | $0.296_{\pm 0.037}$ | $0.648_{\pm 0.027}$ |
| WGAN+RS | $0.794_{\pm 0.020}$ | $0.088_{\pm 0.035}$ | $0.312_{\pm 0.056}$ | $0.617_{\pm 0.018}$ |
| WGAN+DRL | $\mathbf{0.801}_{\pm 0.014}$ | $\mathbf{0.340}_{\pm 0.122}$ | $\mathbf{0.339}_{\pm 0.049}$ | $\mathbf{0.652}_{\pm 0.036}$ |
| TableGAN | $0.659_{\pm 0.035}$ | $\mathbf{0.228}_{\pm 0.035}$ | $0.393_{\pm 0.010}$ | $0.615_{\pm 0.030}$ |
| TableGAN+RS | $0.648_{\pm 0.046}$ | $0.172_{\pm 0.024}$ | $0.389_{\pm 0.015}$ | $0.599_{\pm 0.036}$ |
| TableGAN+DRL | $\mathbf{0.693}_{\pm 0.028}$ | $0.196_{\pm 0.076}$ | $\mathbf{0.401}_{\pm 0.018}$ | $\mathbf{0.628}_{\pm 0.038}$ |
| CTGAN | $0.799_{\pm 0.033}$ | $0.159_{\pm 0.042}$ | $0.379_{\pm 0.061}$ | $0.675_{\pm 0.015}$ |
| CTGAN+RS | $0.795_{\pm 0.014}$ | $0.095_{\pm 0.019}$ | $0.342_{\pm 0.054}$ | $0.650_{\pm 0.009}$ |
| CTGAN+DRL | $\mathbf{0.815}_{\pm 0.011}$ | $\mathbf{0.308}_{\pm 0.118}$ | $\mathbf{0.409}_{\pm 0.007}$ | $\mathbf{0.680}_{\pm 0.011}$ |
| TVAE | $0.802_{\pm 0.012}$ | $0.351_{\pm 0.182}$ | $\mathbf{0.330}_{\pm 0.016}$ | $0.686_{\pm 0.004}$ |
| TVAE+RS | $0.778_{\pm 0.026}$ | $0.061_{\pm 0.010}$ | $0.283_{\pm 0.007}$ | $0.465_{\pm 0.001}$ |
| TVAE+DRL | $\mathbf{0.832}_{\pm 0.014}$ | $\mathbf{0.487}_{\pm 0.096}$ | $\mathbf{0.330}_{\pm 0.014}$ | $\mathbf{0.694}_{\pm 0.006}$ |
| GOGGLE | $0.648_{\pm 0.074}$ | $0.076_{\pm 0.015}$ | $0.296_{\pm 0.066}$ | $0.566_{\pm 0.050}$ |
| GOGGLE+RS | $0.639_{\pm 0.068}$ | $0.084_{\pm 0.023}$ | $\mathbf{0.322}_{\pm 0.065}$ | $0.549_{\pm 0.051}$ |
| GOGGLE+DRL | $\mathbf{0.673}_{\pm 0.039}$ | $\mathbf{0.281}_{\pm 0.139}$ | $0.310_{\pm 0.057}$ | $\mathbf{0.636}_{\pm 0.020}$ |

**Efficacy.** Tables 18, 19, 20 show the efficacy, along with the standard deviations from the mean, for each DGM+LL model and the corresponding DGM+DRL models on every classification dataset, using the F1, $w$F1, and AUC metrics, respectively. Similarly, in Table 21, we report MAE and RMSE for each unconstrained DGM+LL model and the corresponding DGM+DRL models on the regression dataset (i.e., House).

## M  BACKGROUND KNOWLEDGE ALIGNMENT: A QUALITATIVE ANALYSIS

Figure 6 accompanies Figure 4 from the main body of our paper and shows the relevant sample space for the same two constraints from the House dataset: *if the Zipcode is 98004 or 98005 then the Price is greater than 400K USD* and *if the Zipcode is between 98006 and 98008 then the Price exceeds 225K USD*. As we can see, in all cases, the unconstrained DGMs and the DGMs+LL fail to comply with the constraints. Unlike the synthetic data from the unconstrained DGMs, the samples generated using our DRL layer never cross into the areas that mark regions where datapoints violate

Table 13: Efficacy comparison between the unconstrained DGM models and their DRL counterparts in terms of AUC. The performance, along with the standard deviation, is reported for each classification dataset.

| | URL | CC | LCLD | HELOC |
|---|---|---|---|---|
| WGAN | $0.870 \pm 0.012$ | $0.814 \pm 0.072$ | $0.605 \pm 0.010$ | $\mathbf{0.717} \pm 0.021$ |
| WGAN+RS | $0.862 \pm 0.019$ | $0.570 \pm 0.070$ | $0.611 \pm 0.022$ | $0.685 \pm 0.023$ |
| WGAN+DRL | $\mathbf{0.875} \pm 0.007$ | $\mathbf{0.885} \pm 0.050$ | $\mathbf{0.623} \pm 0.023$ | $\mathbf{0.717} \pm 0.029$ |
| TableGAN | $0.843 \pm 0.020$ | $\mathbf{0.802} \pm 0.044$ | $0.655 \pm 0.011$ | $0.707 \pm 0.007$ |
| TableGAN+RS | $0.854 \pm 0.016$ | $0.682 \pm 0.086$ | $0.653 \pm 0.010$ | $0.685 \pm 0.020$ |
| TableGAN+DRL | $\mathbf{0.865} \pm 0.022$ | $0.742 \pm 0.096$ | $\mathbf{0.657} \pm 0.007$ | $\mathbf{0.709} \pm 0.011$ |
| CTGAN | $0.859 \pm 0.040$ | $0.914 \pm 0.039$ | $\mathbf{0.651} \pm 0.020$ | $0.744 \pm 0.009$ |
| CTGAN+RS | $0.856 \pm 0.010$ | $0.515 \pm 0.083$ | $0.615 \pm 0.031$ | $0.706 \pm 0.014$ |
| CTGAN+DRL | $\mathbf{0.883} \pm 0.009$ | $\mathbf{0.955} \pm 0.022$ | $0.643 \pm 0.019$ | $\mathbf{0.745} \pm 0.008$ |
| TVAE | $0.863 \pm 0.011$ | $0.858 \pm 0.100$ | $0.631 \pm 0.004$ | $0.750 \pm 0.004$ |
| TVAE+RS | $0.846 \pm 0.024$ | $0.522 \pm 0.040$ | $0.480 \pm 0.008$ | $0.497 \pm 0.006$ |
| TVAE+DRL | $\mathbf{0.893} \pm 0.010$ | $\mathbf{0.926} \pm 0.039$ | $\mathbf{0.635} \pm 0.002$ | $\mathbf{0.752} \pm 0.003$ |
| GOGGLE | $0.742 \pm 0.071$ | $0.549 \pm 0.051$ | $0.551 \pm 0.034$ | $0.600 \pm 0.056$ |
| GOGGLE+RS | $0.727 \pm 0.060$ | $0.571 \pm 0.077$ | $0.532 \pm 0.049$ | $0.592 \pm 0.052$ |
| GOGGLE+DRL | $\mathbf{0.747} \pm 0.029$ | $\mathbf{0.758} \pm 0.091$ | $\mathbf{0.563} \pm 0.027$ | $\mathbf{0.691} \pm 0.039$ |

Table 14: Efficacy performance comparison between DGM and DGM+DRL models trained on House, using MAE and RMSE.

| | MAE | RMSE |
|---|---|---|
| WGAN | $547652.6 \pm 6.1$ | $688130.1 \pm 4.8$ |
| WGAN+DRL | $\mathbf{547637.5} \pm 17.4$ | $\mathbf{688118.0} \pm 14.9$ |
| TableGAN | $547655.3 \pm 5.9$ | $688132.7 \pm 4.9$ |
| TableGAN+DRL | $\mathbf{547653.6} \pm 22.8$ | $\mathbf{688131.4} \pm 18.7$ |
| CTGAN | $547652.9 \pm 2.7$ | $688130.4 \pm 2.0$ |
| CTGAN+DRL | $\mathbf{547642.9} \pm 18.1$ | $\mathbf{688122.1} \pm 14.0$ |
| TVAE | $547650.0 \pm 5.1$ | $688128.5 \pm 4.2$ |
| TVAE+DRL | $\mathbf{547645.4} \pm 31.8$ | $\mathbf{688124.6} \pm 27.8$ |
| GOGGLE | $547639.5 \pm 13.8$ | $688119.6 \pm 11.5$ |
| GOGGLE+DRL | $\mathbf{547633.9} \pm 16.0$ | $\mathbf{688115.7} \pm 11.3$ |

the constraints and, in addition, their distribution resembles more closely the real data in all five cases.

In addition, we show a similar comparison but for a different dataset and different constraints. Specifically, we consider the following two constraints from the URL dataset: *if the Number of Subdomains is less than 2 then the Hostname length is less than 30* and *if the Number of Subdomains is less than 3 then the Hostname length is less than 55*. The two constraints capture the relation between the two features (i.e., *Number of Subdomains* and *Hostname length*) and, differently from the two constraints from the House dataset mentioned above, their respective violation space intersects, as shown in red in Figure 7. Nevertheless, our constrained models never violate any of the constraints, unlike the unconstrained and DGM+LL models.

Finally, in order to show the unintended consequences of using rejection sampling, we visualise using t-SNE (van der Maaten & Hinton, 2008) the differences between the distributions of the samples generated by the standard DGMs and by DGMs+RS (i.e, with rejection sampling). These visualisations can be found in Figure 8. As we can see in the Figure, there can be cases where rejection sampling actually creates some changes in the distribution, which can then affect the machine learning efficacy.

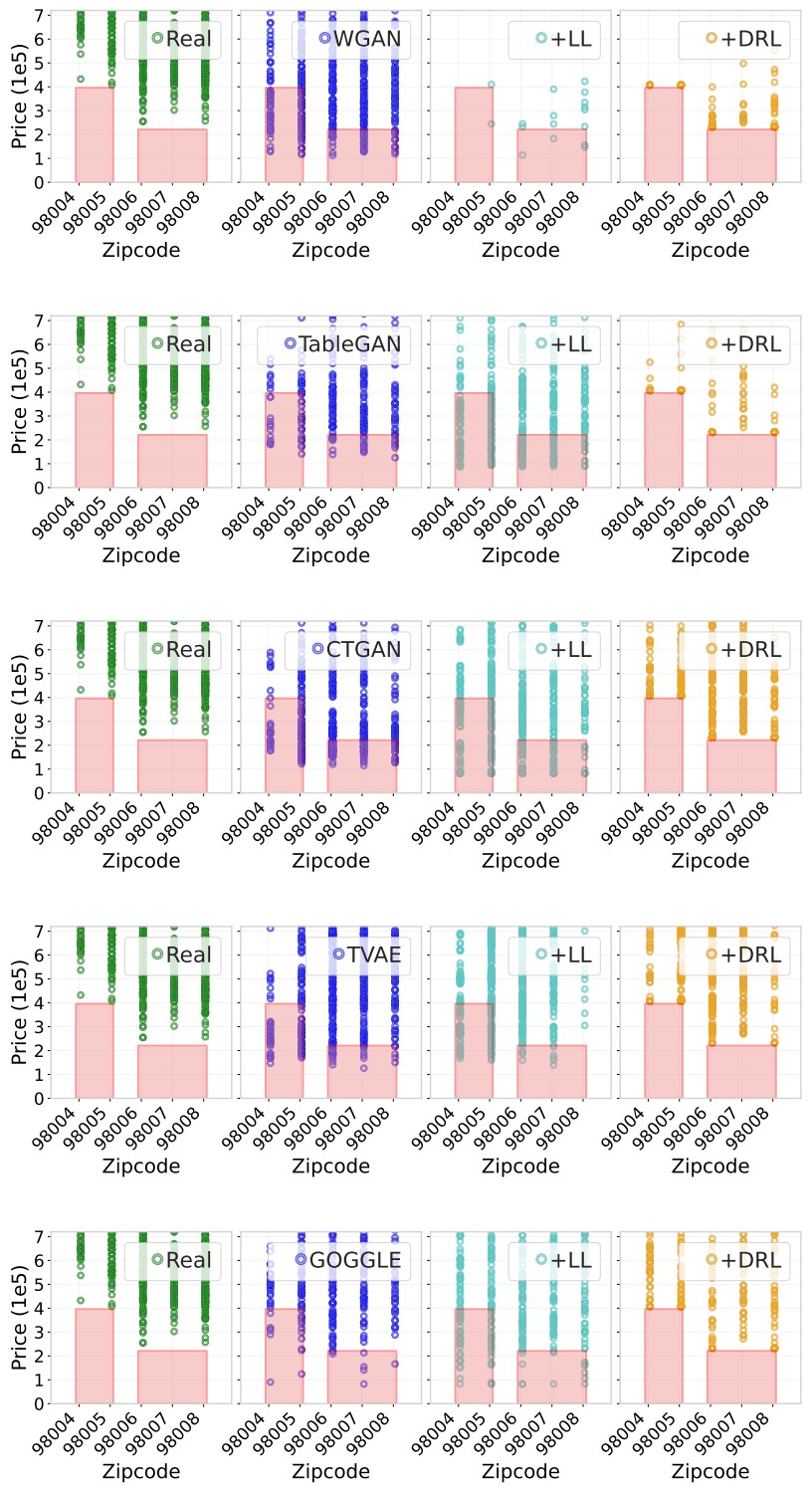

Figure 6: Comparison of sample distributions between real data and synthetic data generated by the unconstrained DGM models and their corresponding DGM+LL and DGM+DRL models, using the *Zipcode* and *Price* features from the House dataset. In order (from the top to the bottom row), the DGM models used in the plots are: WGAN, TableGAN, CTGAN, TVAE, and GOGGLE. The areas in red indicate regions where samples violate the constraints on the given features.

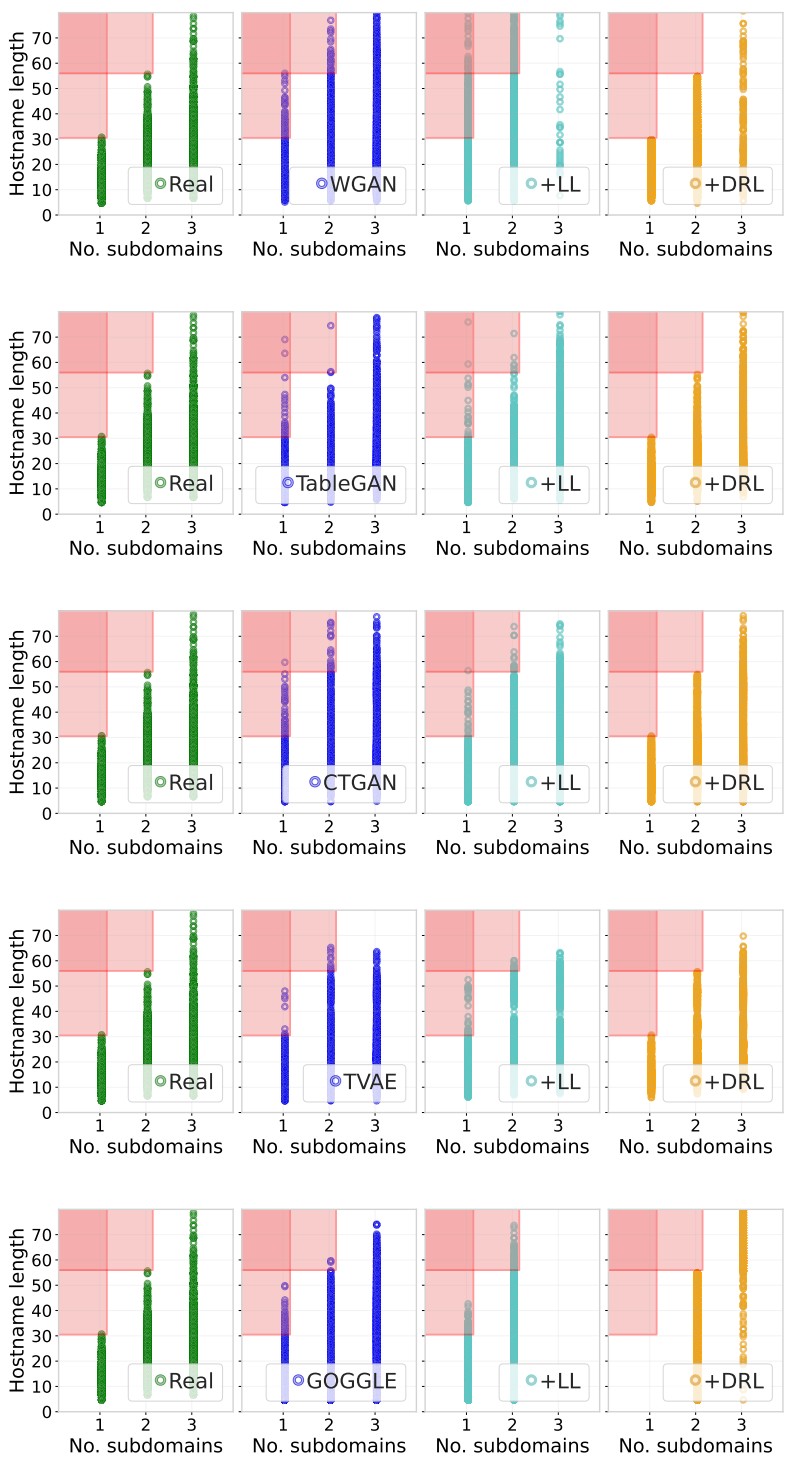

Figure 7: Comparison of sample distributions between real data and synthetic data generated by the unconstrained DGM models and their corresponding DGM+LL and DGM+DRL models, using the *No. subdomains* and *Hostname length* features from the URL dataset. In order (from the top to the bottom row), the DGM models used in the plots are: WGAN, TableGAN, CTGAN, TVAE, and GOGGLE. The areas in red indicate regions where samples violate the constraints on the given features.

Table 15: Constraint violation rate (CVR) for each DGM+LL model and dataset.

| Constraint Type | Model/Dataset | URL | CCS | LCLD | Heloc | House |
|---|---|---|---|---|---|---|
| Linear | All models + LL | **0.0**±0.0 | **0.0**±0.0 | **0.0**±0.0 | **0.0**±0.0 | **0.0** ±0.0 |
| | All models + DRL | **0.0**±0.0 | **0.0**±0.0 | **0.0**±0.0 | **0.0**±0.0 | **0.0** ±0.0 |
| Disjunctive | WGAN+LL | 8.9±3.2 | 51.5±11.2 | 27.0±3.6 | 20.6±6.3 | 100.0±0.0 |
| | TableGAN+LL | 3.6±0.8 | 54.0±17.8 | 11.3±0.9 | 26.6±7.7 | 23.9±2.7 |
| | CTGAN+LL | 7.0±2.6 | 55.7±16.3 | 2.6±1.1 | 2.6±2.4 | 10.8±7.8 |
| | TVAE+LL | 6.8±0.6 | 8.4±2.0 | 5.8±0.8 | 0.0±0.0 | 13.0±12.6 |
| | GOGGLE+LL | 6.5±7.0 | 23.0±10.7 | 81.9±6.5 | 11.5±7.1 | 2.6±2.6 |
| | All models + DRL | **0.0**±0.0 | **0.0**±0.0 | **0.0**±0.0 | **0.0**±0.0 | **0.0** ±0.0 |

Table 16: Samplewise constraints violation coverage (sCVC) for each DGM+LL model and each dataset.

| Constraint Type | Model/Dataset | URL | CCS | LCLD | Heloc | House |
|---|---|---|---|---|---|---|
| Linear | All models + LL | **0.0**±0.0 | **0.0**±0.0 | **0.0**±0.0 | **0.0**±0.0 | **0.0** ±0.0 |
| | All models + DRL | **0.0**±0.0 | **0.0**±0.0 | **0.0**±0.0 | **0.0**±0.0 | **0.0** ±0.0 |
| Disjunctive | WGAN+LL | 0.9±0.4 | 6.9±1.1 | 2.2±0.4 | 4.2±1.4 | 9.1±0.0 |
| | TableGAN+LL | 0.4±0.1 | 6.5±2.3 | 1.0±0.1 | 5.9±1.7 | 2.2±0.3 |
| | CTGAN+LL | 0.7±0.3 | 7.0±2.1 | 0.2±0.1 | 0.5±0.5 | 1.0±0.7 |
| | TVAE+LL | 0.7±0.1 | 0.9±0.2 | 0.4±0.1 | 0.0±0.0 | 1.2±1.2 |
| | GOGGLE+LL | 0.7±0.7 | 2.6±1.4 | 11.2±2.1 | 2.3±1.4 | 0.2±0.2 |
| | All models + DRL | **0.0**±0.0 | **0.0**±0.0 | **0.0**±0.0 | **0.0**±0.0 | **0.0** ±0.0 |

## N  SAMPLE GENERATION TIME

As we can see from Table 22, the DGM+DRL models bring additional time to the sample generation runtimes. However, this is often much smaller than the additional time required to use an unconstrained model and then doing rejection sampling. Indeed, the largest runtime difference registered between a unconstrained DGM and its constrained counterpart is of only 0.12 seconds (i.e., for CTGAN on the URL and LCLD datasets). Notably, in 20 out of 25 cases, the overhead for the DGM+DRL models is less than 0.1 seconds. On the other hand, for the DGM+RS models, the sample generation procedure timed out after 24h for all the models tested on the House dataset (where we had 100% CVR). The registered times are also not very promising for any of the other datasets when using DGM+RS, where the minimum absolute difference registered equals 0.07 seconds and the maximum equals 5.55 seconds (notice that no DGM nor DGM+DRL model has sampling time above 0.29 seconds).

## O  REAL DATA PERFORMANCE

In Table 25 we report the average F1-score, weighted F1-score and Area Under the ROC Curve obtained by training the same six classifiers (resp. four regressors) on the four real classification datasets (resp. real regression dataset). This allows us to compare the machine learning efficacy of the synthetic data with one of the real data. As we can see from the results, the synthetic data generated with all the DGMs (unconstrained, +LL and +DRL) manage to obtain very good results. In multiple occasions, the models trained on the synthetic data not only get results comparable with the ones obtained with the real data, but actually perform better than them. In spite of this, the classification models trained on synthetic data never manage to get better performance than those trained on the real data for all metrics. On the other hand, the regressors trained on the synthetic version of the House dataset always manage to get lower MAE and RMSE, no matter the DGM used to generate the synthetic data (with the exception of TVAE+LL which got slightly higher MAE than the one obtained with the real data).

Table 17: Constraints violation coverage (CVC) for each DGM+LL model and each dataset.

| Constraint Type | Model/Dataset | URL | CCS | LCLD | Heloc | House |
|---|---|---|---|---|---|---|
| Linear | All models + LL | **0.0**±0.0 | **0.0**±0.0 | **0.0**±0.0 | **0.0**±0.0 | **0.0** ±0.0 |
| | All models + DRL | **0.0**±0.0 | **0.0**±0.0 | **0.0**±0.0 | **0.0**±0.0 | **0.0** ±0.0 |
| Disjunctive | WGAN+LL | 50.0±0.0 | 40.0±0.0 | 28.0±1.3 | 79.2±1.8 | 13.1±4.1 |
| | TableGAN+LL | 54.0±4.7 | 30.0±0.0 | 27.3±0.0 | 84.8±11.1 | 36.7±0.8 |
| | CTGAN+LL | 52.8±2.3 | 30.0±0.0 | 25.4±2.7 | 41.6±12.8 | 32.7±5.0 |
| | TVAE+LL | 51.2±1.1 | 29.2±1.8 | 22.6±0.6 | 20.8±14.3 | 32.0±4.6 |
| | GOGGLE+LL | 33.6±12.4 | 27.5±5.0 | 46.2±0.0 | 28.0±16.7 | 17.6±1.0 |
| | All models + DRL | **0.0**±0.0 | **0.0**±0.0 | **0.0**±0.0 | **0.0**±0.0 | **0.0** ±0.0 |

Table 18: Efficacy comparison between the DGM+LL models and their DRL counterparts in terms of F1. The performance, along with the standard deviation, is reported for each classification dataset.

| | URL | CC | LCLD | HELOC |
|---|---|---|---|---|
| WGAN+LL | **0.803**±0.038 | **0.359**±0.096 | 0.183±0.094 | 0.694±0.033 |
| WGAN+DRL | 0.800±0.011 | 0.313±0.127 | **0.197**±0.060 | **0.721**±0.027 |
| TableGAN+LL | 0.612±0.111 | **0.169**±0.044 | 0.232±0.026 | **0.638**±0.061 |
| TableGAN+DRL | **0.619**±0.046 | 0.163±0.079 | **0.269**±0.025 | 0.628±0.083 |
| CTGAN+LL | **0.836**±0.002 | 0.250±0.081 | 0.265±0.040 | 0.729±0.027 |
| CTGAN+DRL | **0.836**±0.004 | **0.288**±0.116 | **0.288**±0.013 | **0.744**±0.020 |
| TVAE+LL | 0.824±0.004 | 0.413±0.057 | 0.158±0.011 | 0.730±0.009 |
| TVAE+DRL | **0.835**±0.009 | **0.467**±0.100 | **0.189**±0.022 | **0.731**±0.009 |
| GOGGLE+LL | **0.787**±0.014 | 0.233±0.180 | 0.284±0.123 | **0.723**±0.018 |
| GOGGLE+DRL | 0.720±0.086 | **0.253**±0.144 | **0.298**±0.153 | 0.698±0.023 |

## P COMPARISON BETWEEN DGMS+DRL AND LLM-BASED TABULAR DATA GENERATION

A recent trend in the tabular data generation field has been to use LLMs to perform the task. While these models are very promising, they are also not exempt from the problems that affect the other DGMs. In this section we thus compare the performance of the standard DGMs equipped with our DRL and GreAT (Borisov et al., 2023), a state-of-the-art LLM-based tabular data generator. For all datasets considered, GreAT was trained and run on an A100 GPU with 40GB of RAM, using the pre-defined hyperparameters.

Table 23: CVR for each model and dataset. Cases with CVR≥50% are underlined. Best results are in bold.

| | URL | CCS | LCLD | Heloc | House |
|---|---|---|---|---|---|
| GreAT | 0.7±0.2 | 98.0±0.3 | 1.1±0.1 | 9.60±0.5 | 15.7±1.4 |
| All + DRL | **0.0**±0.0 | **0.0**±0.0 | **0.0**±0.0 | **0.0**±0.0 | **0.0**±0.0 |

Firstly, in Table 23, we report the CVR obtained with GreAT and with all the models+DRL. As we can see from the Table, even though in two out of five cases GreAT manages to get a very low CVR, for CCS its CVR shoots up to 98.0%. As CCS is by far our smallest dataset (with only 1K datapoints in the training set), this hints to the fact that LLM-based models struggle to learn the distribution from datasets with few datapoints.

Secondly, we generate 1,000 samples with GreAT and we report the average runtime in Table 24 together with the average sampling time obtained with the DGMs+DRL. As we can see from the Table, GreAT takes two orders of magnitude longer to perform the sampling than the average DGM+DRL model.

Table 24: Sample generation time in seconds.

| | URL | CCS | LCLD | Heloc | House |
|---|---|---|---|---|---|
| GreAT | 51.8 | 28.4 | 22.3 | 26.3 | 14.4 |
| DGM+DRL | 0.22 | 0.13 | 0.14 | 0.10 | 0.13 |

Table 19: Efficacy comparison between the DGM+LL models and their DRL counterparts in terms of $w$F1. The performance, along with the standard deviation, is reported for each classification dataset.

|  | URL | CC | LCLD | HELOC |
|---|---|---|---|---|
| WGAN+LL | 0.799±0.022 | **0.383**±0.092 | 0.330±0.068 | **0.662**±0.021 |
| WGAN+DRL | **0.801**±0.014 | 0.340±0.122 | **0.339**±0.049 | 0.652±0.036 |
| TableGAN+LL | **0.695**±0.071 | **0.203**±0.042 | 0.373±0.017 | **0.633**±0.036 |
| TableGAN+DRL | 0.693±0.028 | 0.196±0.076 | **0.401**±0.018 | 0.628±0.038 |
| CTGAN+LL | **0.820**±0.008 | 0.271±0.083 | 0.392±0.030 | **0.688**±0.010 |
| CTGAN+DRL | 0.815±0.011 | **0.308**±0.118 | **0.409**±0.007 | 0.680±0.011 |
| TVAE+LL | 0.816±0.008 | 0.436±0.055 | 0.310±0.011 | 0.691±0.007 |
| TVAE+DRL | **0.832**±0.014 | **0.487**±0.096 | **0.330**±0.014 | **0.694**±0.006 |
| GOGGLE+LL | **0.749**±0.029 | 0.262±0.173 | **0.310**±0.039 | **0.663**±0.012 |
| GOGGLE+DRL | 0.673±0.039 | **0.281**±0.139 | **0.310**±0.057 | 0.636±0.020 |

Table 20: Efficacy comparison between the DGM+LL models and their DRL counterparts in terms of AUC. The performance, along with the standard deviation, is reported for each classification dataset.

|  | URL | CC | LCLD | HELOC |
|---|---|---|---|---|
| WGAN+LL | 0.869±0.014 | 0.857±0.058 | 0.608±0.021 | **0.732**±0.013 |
| WGAN+DRL | **0.875**±0.007 | **0.885**±0.050 | **0.623**±0.023 | 0.717±0.029 |
| TableGAN+LL | **0.868**±0.007 | **0.794**±0.015 | 0.640±0.005 | 0.704±0.030 |
| TableGAN+DRL | 0.865±0.022 | 0.742±0.096 | **0.657**±0.007 | **0.709**±0.011 |
| CTGAN+LL | 0.880±0.007 | **0.959**±0.027 | 0.641±0.015 | **0.755**±0.007 |
| CTGAN+DRL | **0.883**±0.009 | 0.955±0.022 | **0.643**±0.019 | 0.745±0.008 |
| TVAE+LL | 0.878±0.007 | **0.933**±0.036 | 0.633±0.003 | 0.747±0.007 |
| TVAE+DRL | **0.893**±0.010 | 0.926±0.039 | **0.635**±0.002 | **0.752**±0.003 |
| GOGGLE+LL | **0.802**±0.016 | **0.765**±0.084 | 0.554±0.039 | **0.719**±0.005 |
| GOGGLE+DRL | 0.747±0.029 | 0.758±0.091 | **0.563**±0.027 | 0.691±0.039 |

Table 21: Efficacy performance comparison between DGM+LL and DGM+DRL models trained on House, using MAE and RMSE.

|  | MAE | RMSE |
|---|---|---|
| WGAN+LL | 547638.5±11.4 | 688118.2±11.0 |
| WGAN+DRL | **547637.5**±17.4 | **688118.0**±14.9 |
| TableGAN+LL | **547638.0**±18.9 | **688118.3**±17.9 |
| TableGAN+DRL | 547653.6±22.8 | 688131.4±18.7 |
| CTGAN+LL | **547642.3**±14.6 | **688121.4**±14.5 |
| CTGAN+DRL | 547642.9±18.1 | 688122.1±14.0 |
| TVAE+LL | 547658.1±9.8 | 688137.4±9.4 |
| TVAE+DRL | **547645.4**±31.8 | **688124.6**±27.8 |
| GOGGLE+LL | 547651.9±9.9 | 688129.6±8.4 |
| GOGGLE+DRL | **547633.9**±16.0 | **688115.7**±11.3 |

This difference is particularly striking given that GreAT was the only model requiring an A100 to run.

This analysis shows that not only LLM-based are still prone to the violation of the constraints, but also that they require more powerful hardware and much longer time to sample.

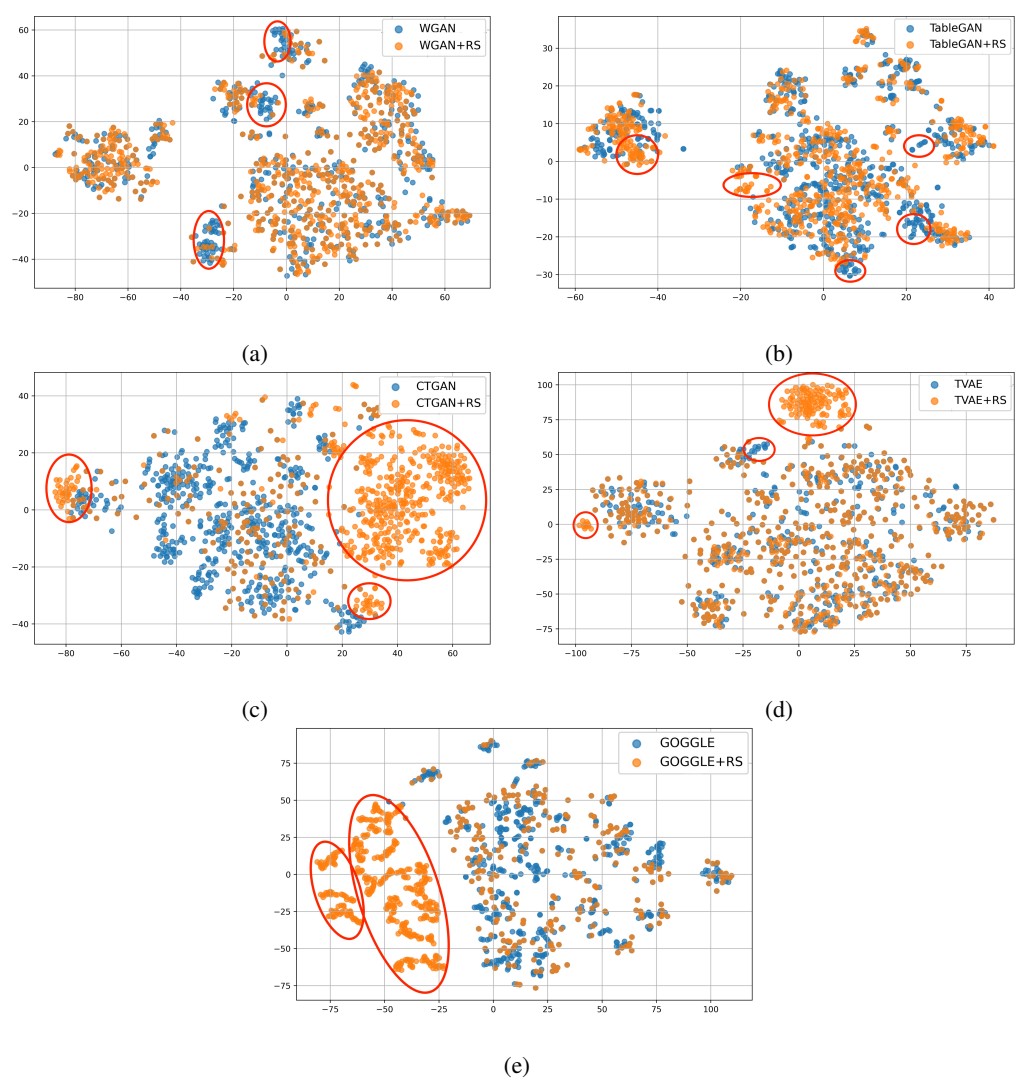

(a)

(b)

(c)

(d)

(e)

Figure 8: t-SNE visualisations of the distribution of the samples generated for the CCS dataset by (i) WGAN and WGAN+RS in Figure 8a, (ii) TableGAN and TableGAN+RS in Figure 8b, (iii) CTGAN and CTGAN+RS in Figure 8c, (iv) TVAE and TVAE+RS in Figure 8d, and (v) GOGGLE and GOGGLE+RS in Figure 8e. Note that changes in distribution are marked with red contours.

Table 22: Sample generation time (in seconds) for all DGMs and their respective DGM+DRL models and DGM+RS models (i.e, DGMs with rejection sampling), for all datasets. The hyphen indicates timeout after 24h.

|  | URL | CCS | LCLD | Heloc | House |
|---|---|---|---|---|---|
| WGAN | 0.01 | 0.01 | 0.01 | 0.01 | 0.00 |
| WGAN+RS | 0.08 | 0.11 | 0.12 | 0.25 | - |
| WGAN+DRL | 0.08 | 0.07 | 0.08 | 0.04 | 0.09 |
| TableGAN | 0.21 | 0.10 | 0.09 | 0.10 | 0.07 |
| TableGAN+RS | 0.43 | 1.57 | 0.38 | 0.78 | - |
| TableGAN+DRL | 0.28 | 0.14 | 0.19 | 0.15 | 0.18 |
| CTGAN | 0.16 | 0.11 | 0.10 | 0.09 | 0.07 |
| CTGAN+RS | 0.45 | 1.71 | 0.37 | 1.01 | - |
| CTGAN+DRL | 0.28 | 0.19 | 0.22 | 0.14 | 0.16 |
| TVAE | 0.14 | 0.08 | 0.07 | 0.06 | 0.05 |
| TVAE+RS | 0.37 | 0.31 | 1.08 | 0.96 | - |
| TVAE+DRL | 0.25 | 0.16 | 0.20 | 0.11 | 0.14 |
| GOGGLE | 0.22 | 0.08 | 0.08 | 0.06 | 0.06 |
| GOGGLE+RS | 0.48 | 0.35 | 5.63 | 0.25 | - |
| GOGGLE+DRL | 0.29 | 0.14 | 0.11 | 0.09 | 0.16 |

Table 25: Efficacy scores calculated on real data. For classification datasets URL, CCS and LCLD, we used F1-score, weighted F1-score and Area Under the Curve to measure the performance, while for the regression dataset House, we used the Mean Absolute Error metrics and the Root Mean Square Error.

|  | F1 | $w$F1 | AUC | MAE | RMSE |
|---|---|---|---|---|---|
| URL | $0.884_{\pm 0.007}$ | $0.875_{\pm 0.014}$ | $0.903_{\pm 0.009}$ | - | - |
| CCS | $0.529_{\pm 0.011}$ | $0.547_{\pm 0.011}$ | $0.948_{\pm 0.010}$ | | |
| LCLD | $0.171_{\pm 0.030}$ | $0.316_{\pm 0.013}$ | $0.645_{\pm 0.007}$ | - | - |
| Heloc | $0.772_{\pm 0.003}$ | $0.662_{\pm 0.011}$ | $0.707_{\pm 0.008}$ | - | - |
| House | - | - | - | $547655.7_{\pm 38.2}$ | $688133.1_{\pm 30.8}$ |

