# OpenReview forum: "Beyond the convexity assumption: Realistic tabular data generation under quantifier-free real linear constraints"
_ICLR.cc/2025/Conference — ICLR 2025 Poster_

### Official Review · Reviewer_KTDt · 2024-11-02

**Soundness:** 2
**Presentation:** 2
**Contribution:** 2
**Rating:** 5
**Confidence:** 4

**Summary:**

This paper proposes a novel Disjunctive Refinement Layer added to neural network synthetic data generators (to replace linear constraint layer) that naturally enforces complex (including non-convex and disconnected) QFLRA constraints on tabular data without rejection sampling, and also improves the synthetic data's utility.

**Strengths:**

+ It uses a disjunction of linear inequalities to decompose complex non-convex and disconnected constraints into a combination of simple constraints, whose satisfiability implies the entire inequality system's satisfiability.
+ It derives and resolves implicit constraints among previous variables that ensure the satisfiability of constraints on the latter variable, and hence allows satisfiability in the entire system.

**Weaknesses:**

- Mathematical statements need clarification:

    1. Line 127: If $w_k=0,\forall k$ and $b < 0$, which satisfies your constraint that $w_k,b\in\mathbb{R}$, would the claim that $\Omega(\Phi)$ is non-empty still be true? (Although similar case is addressed in Line 275-277, it does not validate Line 127.)
    2. Line 170-172: What if $\Psi$s in $\Pi$ are not simplified and somewhat standardized?
            1) If for some $\Psi$, $l^\Psi_i >= r^\Psi_i$ (violating the constraint "whenever ..." in Line 159, if it could be violated), would the definitions of $l^\Pi_i$ and $r^\Pi_i$ still be valid?
            2) If for some $\Psi,\Psi'\in\Pi$, $[l^\Psi_i, r^\Psi_i]\cap$[l^{\Psi'}_i,r^{\Psi'}_i]\ne\emptyset$ (so that less than $|\Pi|+1$ disjoint intervals are created), would the definitions of $l^\Pi_i$ and $r^\Pi_i$ still be valid?
    3. Line 259-260: Ambiguous (although can be guessed from textual explanation), it's $(A\setminus B)\cup C$ or $A\setminus(B\cup C)$?
    4. Line 285-287: Similar problem as in Line 170-172

- Some implementation details not mentioned in paper and appendix:

    1. How the DRL is applied in all the experimented models is not explicitly stated, including Appendix.
    2. How constraints are determined (or if manually determined, what are they) is not mentioned in the main content nor appendix.

- The paper claims the additional sampling time is negligible, but I would like to doubt this on the following basis:

    1. 1000 samples could be a bit too small, taking the shared overhead of initializing the task and data transformation into account.
    2. The sampling time is expected to be approximately in linear correlation to the number of samples, so that ~0.1s for 1000 samples mean ~1s for 10000 samples, ~10s for 100000 samples, etc. What matters more is not the absolute additional time per 1000 samples, but the relative additional time (e.g. how many times more needed). According to the reported generation time in Table 5, in 3/5 cases DRL needs more than twice the time than non-DRL. This might not be negligible.
    3. DRL is also added during training, given that it could need ~2x times the time to generate samples, it naturally leads to the question that does it need also ~2x times to train.

- Although the method is novel and sound, the utility of the proposed method is doubtful:

    1. On tabular data generation literature, a recently influential trend of the use of language models and large language models is not discussed, while they are better-performing than GANs, likely to have a very low CVR (may not be 0% in theory but could be very close to it in practice). If these recent models naturally achieves near-perfect result on constraints, then the proposed method may not be useful in practice. Thus, it needs to be discussed.
    2. The authors did not compare the proposed method versus direct rejection sampling. Rejection sampling also guarantees 100% satisfaction of all constraints, and according to the reported timing in Section 4.3 (Table 5), it is not necessarily worse. e.g. for URL dataset, non-WGAN models has a CVR of about 10%. This means ~0.15/0.9=0.17s < 0.27s is needed, so it's even faster. Moreover, rejection sampling preserves naturally generated samples from generators instead of editing them, and there may be a chance that its utility can be even better.
    3. Putting the above two points together, there could be a model with <1% violation rate, so <+1% time for generation, that obtains better result than +DRL in the older models.


- A serious issue is that the page limit is 10 but section 7 and 8 in this paper is in the page 11, which can potentially lead to desk reject.

**Questions:**

As mentioned in Weaknesses.

---

> ### Author Response · Authors · 2024-11-20
>
> We thank the reviewer for their appreciation of the hardness of the problem we are handling. In the new version of the PDF we have highlighted all the new text related to this review in *blue*.
>
> Below we provide point-by-point answers to all the questions.
>
> ***Mathematical Statements:***
>
> Q1. *Line 127: If $w_k=0$ and $b <0$, which satisfies your constraint that $w_k, b \in \mathbb{R}$, would the claim that $\Omega(\Phi)$ is non-empty still be true? (Although similar case is addressed in Line 275-277, it does not validate Line 127.)*
>
> A1. The claim that $\Omega(\Phi)$ is not empty holds if we assume that at least a variable appears in $\Phi$ which is the assumption under which wrote those properties (as specified directly below at line 131 of the original paper).
>
> Q2. *Line 170-172: What if $\Psi$s in $\Pi$  are not simplified and somewhat standardized? 1) If for some $\Psi$, $l_i^{\Psi} \ge r_i^{\Psi}$ (violating the constraint "whenever ..." in Line 159, if it could be violated), would the definitions of $l_i^{\Pi}$  and $r_i^{\Pi}$ still be valid? 2) If for some $\Psi, \Psi’ \in \Pi$, $ [l^{\Psi}_i,r^{\Psi'}_i] \cup [l^{\Psi'}_i,r^{\Psi'}_i] \neq \emptyset$, would the definitions of $l^\Pi_i$ and $r^\Pi_i$ still be valid?*
>
> A2. The definitions of $l_i^{\Pi}$ and $r_i^{\Pi}$ are absolutely valid even if there exist $l_i^{\Psi} \ge r_i^{\Psi}$. Lemma 3.1 shows that, for *every*  constraint, $DRL(\tilde{x})$ satisfies $\Pi$ and is optimal w.r.t. $\tilde x$. Notice that these boundaries are of interest only when $\tilde x$  violates the constraints; if the original sample does not violate the constraints then $DRL(\tilde{x})$ leaves the sample unchanged. 2) $l_i^{\Pi}$ and $r_i^{\Pi}$ are absolutely valid even when $[l_i^{\Psi}, r_i^{\Psi}] \cap [l_i^{\Psi’}, r_i^{\Psi’}] \neq \emptyset$ for the same reasons as above.
>
> Q3. *Line 259-260: Ambiguous (although can be guessed from textual explanation)*
>
> A3. We thank the reviewer for pointing this out. We have added the parentheses to improve readability. As explained in the text, it is indeed $(A \setminus B) \cup C$.
>
> Q4. *Line 285-287: Similar problem as in Line 170-172*
>
> A4. Yes, for the same reasons as point 2.
>
>
> ***Implementation details not mentioned:***
>
> Q1. *How the DRL is applied in all the experimented models is not explicitly stated, including Appendix.*
>
> A1. We included a diagram for each model to exemplify how the layer should be added in the topology of each of the considered models in Appendix A.
>
> Q2. *How constraints are determined (or if manually determined, what are they) is not mentioned in the main content nor appendix*
>
> A2. Some of the constraints were already included in the original datasets. This is true for URL, Heloc and LCLD. Regarding CCS and House, we manually annotated the constraints using our background knowledge about the problem, then we checked whether the data were compliant with our constraints and finally we retained only those constraints that were satisfied by **all** the datapoints. Simple examples of these constraints (from CCS) state simple facts like "if the feature capturing the number of cigarettes packs per day is greater than 0 then the feature capturing whether the patient smokes or not should be equal to 1" or "the feature representing the age should always be higher than the age at the first intercourse".

---

> > ### Author Response · Authors · 2024-11-20
> >
> > ***Negligible sampling time:***
> >
> > Q1. *1000 samples could be a bit too small, taking the shared overhead of initializing the task and data transformation into account.*
> >
> > A1. We took 1000 samples to give an idea of what the generation time would be. As the reviewer mentioned in Q2 there is approximately a linear correlation between the number of samples generated and the amount of time taken to generate it.
> >
> > Q2. *The sampling time is expected to be approximately in linear correlation to the number of samples, so that ~0.1s for 1000 samples mean ~1s for 10000 samples, ~10s for 100000 samples, etc. What matters more is not the absolute additional time per 1000 samples, but the relative additional time (e.g. how many times more needed). According to the reported generation time in Table 5, in 3/5 cases DRL needs more than twice the time than non-DRL. This might not be negligible.*
> >
> > A2. We thank the reviewer for this comment as we agree with them that our “Sample Generation Time” section might be misread, however, please notice that we included the sampling times for transparency’s sake and we had clearly stated that we expected the models+DRL to be slower. To make it even more transparent, we have modified it: we removed both the comparison between absolute sampling generation times, and the word “negligible”. Moreover, we emphasised the point that, in presence of a high violation rate, DRL introduces a much smaller overhead than doing rejection sampling, which is also further proven by our experiments shown below. It is also worth mentioning that, in order to give a "worst-case scenario estimate”, in the original paper we tested the generation time of the untrained models, which obviously generate many more non-compliant samples and require many more corrections to take place (hence longer time). To demonstrate that the overhead is smaller in normal utilisation scenarios, below we report the sampling times obtained with a fully trained model.
> >
> > In order to ground our claims, we took two models (WGAN and TVAE), making sure to include the model that had the lowest sampling time (i.e., WGAN), and computed the generation time for 1000 samples for:
> >
> > 1. The unconstrained models (WGAN and TVAE)
> > 2. The unconstrained models with rejection sampling (WGAN+RS and TVAE+RS)
> > 3. The untrained models + DRL (WGAN+DRL - epoch 0 and TVAE+DRL - epoch 0)
> > 4. The fully trained models + DRL (WGAN+DRL - last epoch and TVAE+DRL - last epoch).
> >
> > We provide the results in the table below.
> > Please note that we took only two models to make sure we could give a timely response to the reviewer and start the discussion as soon as possible. We have already added these results in Appendix N, Table 22 (page 28). We will keep adding more results as we get to run them.
> >
> > Runtime (in seconds) for each dataset. The worst sampling times are *slanted*.
> >
> > | Model                    | URL   | CCS   | LCLD  | Heloc | House                |
> > |--------------------------|-------|-------|-------|-------|----------------------|
> > | WGAN                     | 0.01  | 0.01  | 0.01  | 0.01  | 0.00                |
> > | WGAN+RS                  | *0.08*  | *0.11*  | *0.12*  | *0.25*  | *timeout after 24h*    |
> > | WGAN+DRL - epoch 0       | *0.08*  | 0.07  | 0.08  | 0.04  | 0.09                |
> > | WGAN+DRL - last epoch    | 0.06  | 0.04  | 0.05  | 0.03  | 0.05                |
> > |
> > | TVAE                     | 0.14  | 0.08  | 0.07  | 0.06  | 0.05                |
> > | TVAE+RS                  | *0.37*  | *0.31*  | *1.08*  | *0.96*  | *timeout after 24h*    |
> > | TVAE+DRL - epoch 0       | 0.25  | 0.16  | 0.20  | 0.11  | 0.14                |
> > | TVAE+DRL - last epoch    | 0.20  | 0.14  | 0.13  | 0.09  | 0.10                |

---

> > > ### Author Response · Authors · 2024-11-20
> > >
> > > From the table, we can see that:
> > >
> > > 1. For the House dataset, the procedure did not terminate even within 24 hours (as expected with 100% violation rate).
> > >
> > > 2. For the Heloc dataset (for which the violation rate was of 80.6% for WGAN and of 44.9% for TVAE), we have that WGAN+RS takes ~25x the time taken by the standard WGAN, ~6x the time taken by WGAN+DRL (epoch 0) and ~8x the time taken by WGAN+DRL (last epoch), and that TVAE+RS takes ~16x the time taken by the standard TVAE, ~8.7x the time taken by TVAE+DRL (epoch 0) and ~10.7x the time taken by TVAE+DRL (last epoch).
> > >
> > > 3. For CCS and LCLD (for which the violation rate was around 45% for WGAN and between 10.3% and 16.9% for TVAE) we have that WGAN+RS takes ~11x longer than the unconstrained WGAN model, ~1.5x longer than WGAN+DRL (epoch 0) and ~2.5x longer than WGAN+DRL (last epoch), and that TVAE+RS takes ~3.8x and ~15.4 longer than the unconstrained TVAE model for CCS and LCLD, respectively,  ~1.9x and ~5.4x longer than TVAE+DRL (epoch 0) for CCS and LCLD, respectively, and ~2.2x and ~8.3x longer than TVAE+DRL (last epoch) for CCS and LCLD, respectively.
> > >
> > > 4. Finally, only for URL (where the violation rate was as little as 22.8% and 10.3% for WGAN and TVAE, respectively) we have that WGAN+RS takes as long as WGAN+DRL (epoch 0) but longer than WGAN+DRL (last epoch), and that TVAE+RS takes longer than both TVAE+DRL (epoch 0) and TVAE+DRL (last epoch) (almost 2x).
> > >
> > > Finally, please note that DRL performs the same operations for all models. Thus, by fixing the dataset, DRL adds the same overhead to each model. Hence, by especially selecting WGAN (i.e., the model with the lowest sampling time) we are maximising the difference between the unconstrained model and WGAN+DRL. Additionally, we are effectively computing the sampling time for the rejection sampling algorithm in the best case scenario, we expect these times to be even higher when using rejection sampling with the other models.
> > >
> > > Q3. *DRL is also added during training, given that it could need ~2x times the time to generate samples, it naturally leads to the question that does it need also ~2x times to train*
> > >
> > > A3. It is absolutely true that it takes longer to train with DRL. However, once the model is trained only the sampling time matters. Also, consider that DRL can be simply added as a post-processing step to any model. We did not want to stress on this point as [1] had already shown that adding a layer at post-processing time is not only possible but also has a very small positive impact on the machine learning efficacy while guaranteeing the satisfaction of the constraints.
> > >
> > >
> > >
> > > ***Utility of the method***
> > >
> > > Q1. *On tabular data generation literature, a recently influential trend of the use of language models and large language models is not discussed, while they are better-performing than GANs, likely to have a very low CVR (may not be 0% in theory but could be very close to it in practice). If these recent models naturally achieves near-perfect result on constraints, then the proposed method may not be useful in practice. Thus, it needs to be discussed.*
> > >
> > > A1. We thank the reviewer for this suggestion. We will add to our related work an overview of the following LLM-based models: [2,3,4,5]. Please let us know if there are any other methods relevant to this discussion. We have also tested the hypothesis that LLM-based models can get near-perfect results on constraints by testing the LLM-based GreaT model [2] on our datasets. In the table below, we report  the average CVR (constraint violation rate, i.e., percentage of datapoints violating at least one constraint).
> > >
> > > |  | URL   | CCS   | LCLD  | Heloc  | House  |
> > > |-------|-------|-------|-------|--------|--------|
> > > | CVR   | 0.68% | 98.0% | 1.1%  | 9.60%  | 15.7%  |
> > >
> > > From the table, it can clearly be seen that the LLMs do not always achieve near-perfect scores. We also tested the efficacy of the synthetic data generated for the dataset that had the lowest violation rate (i.e., the URL dataset) and we got the following results: 0.760, 0.798 and 0.872 in terms of F1, wF1 and AUC, respectively. As we can see, even when CVR is less than 1%, GreAT has an efficacy performance 3.4% lower than the standard WGAN in terms of F1 and similar performance in terms of wF1 and AUC (and even lower when compared to models like CTGAN and TVAE).
> > >
> > > Further, as expected, there seems to be a strong correlation between the ability of the LLMs to learn the underlying data distribution and the number of datapoints present in the original dataset. Indeed, we get a very high CVR for CCS which is our smallest dataset with only 1000 datapoints. As generation methods are often used for small datasets (even simply for data augmentation), there are clearly many scenarios where our method is needed.

---

> > > ### Comment · Reviewer_KTDt · 2024-11-22
> > >
> > > Thanks for authors' responses.
> > >
> > > mathematical statements:
> > > Got it, thanks for the clarification.
> > >
> > > implementation details:
> > > Q1. Nice diagrams. Thanks.
> > > Q2. Is the information of how you constructed constraints for CCS and House present in the paper (including appendix)? it'll be good to have it if it is not.
> > >
> > > Negligible sampling time:
> > > My whole point is to ask for a consideration of the problem. Your explanation and additional information is good.
> > >
> > > Utility of the method:
> > > Same point above, just asking these things to be taken into consideration. I highly suggest you to include this CVR and timing report of GReaT in your paper (or appendix).
> > >
> > > Maybe I have missed it, did you provide any support for your claim "(rejection sampling) also changes the distribution of the generated datapoints often leading to much worse machine learning efficacy"?

---

> > > > ### Author Response · Authors · 2024-11-23
> > > >
> > > > First of all, we would like to thank the reviewer for engaging with our answer in a timely manner and for their comments. We believe that, as a result of this discussion, our paper has become much more comprehensive.
> > > >
> > > > ***Implementation details Q2:***
> > > >
> > > > *Is the information of how you constructed constraints for CCS and House present in the paper (including appendix)? it'll be good to have it if it is not.*
> > > >
> > > > The reviewer is totally right. We have added this information in Appendix G, where we also provided the description of the datasets.
> > > >
> > > > ***Negligible sampling time:***
> > > >
> > > > *My whole point is to ask for a consideration of the problem. Your explanation and additional information is good.*
> > > >
> > > > We thank the reviewer for this point and for their appreciation of our explanation and additional information. As they can see, we have started updating Section 4.3, Appendix N and Table 22 in the Appendix in the light of our discussion and we will keep updating the paper as we have more results. This should give the reader a good overview of the impact that our method has on the sample generation time (also compared to rejection sampling).
> > > >
> > > > ***Utility of the method:***
> > > >
> > > > Q1. *Same point above, just asking these things to be taken into consideration. I highly suggest you to include this CVR and timing report of GReaT in your paper (or appendix).*
> > > >
> > > > As there was no space left in the paper, we created Appendix P, where we included GreAT’s CVR and sampling time.
> > > >
> > > >
> > > > Q2. *Maybe I have missed it, did you provide any support for your claim "(rejection sampling) also changes the distribution of the generated datapoints often leading to much worse machine learning efficacy"?*
> > > >
> > > > In the response above we have included the machine learning efficacy results when performing rejection sampling with WGAN and TVAE (and, as we believe this is a very useful analysis for the reader, once we will have the results for all the models we will include them in the paper as well). From those tables, we can see that in most cases WGAN+RS and TVAE+RS perform worse than their unconstrained counterparts without any rejection sampling. Whenever the efficacy drops, it indicates that models trained on the synthetic data struggle to generalise to the test dataset, which consists of real data points. This suggests that the distribution of data points generated with rejection sampling deviates more from the real distribution than the distribution of synthetic data points generated without rejection sampling. To further show the difference in distribution, we considered the CCS dataset and visualised the distributions of samples  generated by WGAN and WGAN+RS and by TVAE and TVAE+RS using t-SNE in Figure 8 of the paper. There we have also highlighted with red circles the main differences between the distributions. We will include plots for the remaining models in case of acceptance, and we will also try to do so during the discussion period if time permits.

---

> > > > > ### Author Response · Authors · 2024-11-28
> > > > >
> > > > > Dear Reviewer KTDt, first of all, we thank you again for your engagement in the discussion thus far and for your comments. As the PDF update deadline approaches, we would like to provide an overview of the additions we have made.
> > > > >
> > > > > As promised, we continued the analysis on how rejection sampling (RS) impacts the performance of the models and updated our results accordingly.
> > > > > We incorporated the new DGM+RS machine learning efficacy results into the main text (in Table 2), as we believe this analysis would be very useful for the reader. Additionally, we updated: (i) Tables 11, 12 and 13 in Appendix K, where we included the standard deviations for the results of the DGM+RS models, (ii) the t-SNE visualisations for the new DGM+RS models in Figure 8 of Appendix M, and (iii) the sample generation runtime for the DGM+RS models in Table 22 in Appendix N.
> > > > >
> > > > > We completed the RS analysis for 4 out of 5 models and, as seen in Table 2, a clear trend emerges: in 14, 15 and 14 cases (out of 16 cases where RS managed to output a sample set compliant with the constraints within a runtime limit of 24h), RS decreased performance compared to the unconstrained DGMs w.r.t. F1-score, weighted F1-score, and AUC, respectively. We will add the results for the remaining model (i.e., GOGGLE+RS) in case of acceptance.

---

> ### Author Response · Authors · 2024-11-20
>
> Q2. *The authors did not compare the proposed method versus direct rejection sampling. Rejection sampling also guarantees 100% satisfaction of all constraints, and according to the reported timing in Section 4.3 (Table 5), it is not necessarily worse. e.g. for URL dataset, non-WGAN models has a CVR of about 10%. This means ~0.15/0.9=0.17s < 0.27s is needed, so it's even faster. Moreover, rejection sampling preserves naturally generated samples from generators instead of editing them, and there may be a chance that its utility can be even better.*
>
> A2. Again, we would like to thank the reviewer for this comment, which allowed us to directly compare our method with rejection sampling and show that using DRL is much better not only in terms of sampling time but also in terms of machine learning efficacy. We have already shown in the Table above that rejection sampling actually always takes longer than applying DRL. Moreover, we tested whether applying rejection sampling improves or deteriorates the machine learning efficacy. We report the results on our classification datasets (hence F1-score, weighted F1-score and Area under the Curve) in the Tables below. Notice that we could not report the results for the House dataset as the process timed out after 24 hours for rejection sampling. In each table we have highlighted in bold the best result and in red the worst result.
>
> Efficacy in terms of F1-score (F1)
>
>
> | Model      | URL            | CCS            | LCLD           | Heloc          |
> |------------|----------------|----------------|----------------|----------------|
> | WGAN       | 0.794 $\pm$ 0.041 | 0.303 $\pm$ 0.060 | *0.139 $\pm$ 0.053* | 0.665 $\pm$ 0.050 |
> | WGAN+RS    | *0.792 $\pm$ 0.031* | *0.051 $\pm$ 0.037* | 0.156 $\pm$ 0.074 | *0.628 $\pm$ 0.043* |
> | WGAN+DRL   | **0.800 $\pm$ 0.011** | **0.313 $\pm$ 0.127** | **0.197 $\pm$ 0.060** | **0.721 $\pm$ 0.027** |
> ||
> | TVAE       | 0.810 $\pm$ 0.008 | 0.325 $\pm$ 0.190 | *0.185 $\pm$ 0.021* | 0.717 $\pm$ 0.013 |
> | TVAE+RS    | *0.788 $\pm$ 0.023* | *0.024 $\pm$ 0.011* | **0.237 $\pm$ 0.018** | *0.420 $\pm$ 0.007* |
> | TVAE+DRL   | **0.835 $\pm$ 0.009** | **0.467 $\pm$ 0.100** | 0.189 $\pm$ 0.022 | **0.731 $\pm$ 0.009** |
>
> Efficacy in terms of weighted F1-score (wF1)
>
> | Model      | URL            | CCS            | LCLD           | Heloc          |
> |------------|----------------|----------------|----------------|----------------|
> | WGAN       | 0.796 $\pm$ 0.026 | 0.330 $\pm$ 0.057 | *0.296 $\pm$ 0.037* | 0.648 $\pm$ 0.027 |
> | WGAN+RS    | *0.794 $\pm$ 0.020* | *0.088 $\pm$ 0.035* | 0.312 $\pm$ 0.056 | *0.617 $\pm$ 0.018* |
> | WGAN+DRL   | **0.801 $\pm$ 0.014** | **0.340 $\pm$ 0.122** | **0.339 $\pm$ 0.049** | **0.652 $\pm$ 0.036** |
> ||
> | TVAE       | 0.802 $\pm$ 0.012 | 0.351 $\pm$ 0.182 | **0.330 $\pm$ 0.016** | 0.686 $\pm$ 0.004 |
> | TVAE+RS    | *0.778 $\pm$ 0.026* | *0.061 $\pm$ 0.010* | *0.283 $\pm$ 0.007* | *0.465 $\pm$ 0.001* |
> | TVAE+DRL   | **0.832 $\pm$ 0.014** | **0.487 $\pm$ 0.096** | **0.330 $\pm$ 0.014** | **0.694 $\pm$ 0.006** |
>
> Efficacy in terms of Area under the Curve (AUC)
>
> | Model      | URL            | CCS            | LCLD           | Heloc          |
> |------------|----------------|----------------|----------------|----------------|
> | WGAN       | 0.870 $\pm$ 0.012 | 0.814 $\pm$ 0.072 | *0.605 $\pm$ 0.010* | **0.717 $\pm$ 0.021** |
> | WGAN+RS    | *0.862 $\pm$ 0.019* | *0.570 $\pm$ 0.070* | 0.611 $\pm$ 0.022 | *0.685 $\pm$ 0.023* |
> | WGAN+DRL   | **0.875 $\pm$ 0.007** | **0.885 $\pm$ 0.050** | **0.623 $\pm$ 0.023** | **0.717 $\pm$ 0.029** |
> ||
> | TVAE       | 0.863 $\pm$ 0.011 | 0.858 $\pm$ 0.100 | 0.631 $\pm$ 0.004 | 0.750 $\pm$ 0.004 |
> | TVAE+RS    | *0.846 $\pm$ 0.024* | *0.522 $\pm$ 0.040* | *0.480 $\pm$ 0.008* | *0.497 $\pm$ 0.006* |
> | TVAE+DRL   | **0.893 $\pm$ 0.010** | **0.926 $\pm$ 0.039** | **0.635 $\pm$ 0.002** | **0.752 $\pm$ 0.003** |
>
>
> As it can be seen from the tables, WGAN+RS **never** yields higher Efficacy than WGAN+DRL. Similarly, except for one case (for TVAE, on the LCLD dataset, using F1 metric), TVAE+RS yields a lower Efficacy than TVAE+DRL in all other cases. Further, out of the cases where rejection sampling managed to produce a sample set without timing out, 20 out of 24 such cases showed a decrease in efficacy compared to the baseline unconstrained model. We registered the largest decrease for CCS: w.r.t. WGAN, the decrease was equal to 25.2% (24.2% and 24.4%, respectively) using the F1 metric (wF1 and AUC, respectively), while w.r.t. TVAE the decrease was equal to 30.1% (29% and 33.6%, respectively), using the F1 metric (wF1 and AUC, respectively).
>
> These results clearly indicate that adding rejection sampling can indeed shift the learnt distribution, hence confirming that rejection sampling is often not a viable option, and surely in these cases using DRL should be preferred.

---

> > ### Author Response · Authors · 2024-11-20
> >
> > Q3. *Putting the above two points together, there could be a model with <1% violation rate, so <+1% time for generation, that obtains better result than +DRL in the older models.*
> >
> > While we understand the reasoning of the reviewer, this point is automatically invalidated by the fact that both the premises turned out to be false:
> >
> > 1. LLM-based models do not get <1% violation rate.
> >
> > 2. While rejection sampling guarantees the satisfaction of the constraints, it also changes the distribution of the generated datapoints often leading to much worse machine learning efficacy.
> >
> > Finally, notice that in order to train a LLM-based model in time for the rebuttal, we had to get access to A100 GPUs (which were not needed for the other models used in our paper). And, even with such powerful GPUs, we obtained much higher training and sampling times than all the other methods (with and without DRL):
> >
> > | Metric                            | URL   | CCS   | LCLD  | Heloc | House  |
> > |-----------------------------------|-------|-------|-------|-------|--------|
> > | Sampling time (1000 samples, on A100) | 51.8s | 28.4s | 22.3s | 26.3s | 14.4s |
> >
> > Please notice that the sampling time of GreAT is *2 orders of magnitude* higher than both the standard models and the standard models (at epoch 0) + DRL.
> >
> > To conclude, LLM-based models still commit a high number of violations (especially on small datasets) and rejection sampling can often hinder the machine learning efficacy of the synthetic dataset, this thus provides unequivocal demonstration that *our method is very much needed in practice*.
> >
> > ***Page limit***
> >
> > Q1. *A serious issue is that the page limit is 10 but section 7 and 8 in this paper is in the page 11, which can potentially lead to desk reject.*
> >
> > A1. Sections 7 and 8 contain the “Ethics Statement” and the “Reproducibility Statement”, respectively. As stated in the Author Guide (https://iclr.cc/Conferences/2025/AuthorGuide) neither counts towards the page limit. Citing from the webpage above:
> >
> > a. The optional ethic statement will not count toward the page limit, but should not be more than 1 page.
> >
> > b. This optional reproducibility statement will not count toward the page limit, but should not be more than 1 page.
> >
> > ***References***
> >
> > [1] Stoian et al. How Realistic Is Your Synthetic Data? Constraining Deep Generative Models for Tabular Data, ICLR, 2024
> >
> > [2] Borisov et al. Language models are realistic tabular data generators. In The Eleventh International Conference on Learning Representations, 2023. URL https://openreview.net/forum?id= cEygmQNOeI.
> >
> > [3] A. V. Solatorio and O. Dupriez. REaLTabFormer: Generating realistic relational and tabular data using transformers, 2023. URL https://arxiv.org/abs/2302.02041.
> >
> > [4] M. Gulati and P. Roysdon. TabMT: Generating tabular data with masked transformers. In A. Oh, T. Naumann, A. Globerson, K. Saenko, M. Hardt, and S. Levine (eds.), Advances in Neural Information Processing Systems, volume 36, pp. 46245–46254. Curran Associates, Inc., 2023. URL https://proceedings.neurips.cc/paper_files/paper/2023/ file/90debc7cedb5cac83145fc8d18378dc5-Paper-Conference.pdf.
> >
> > [5] Z. Zhao et al. CTAB-GAN+: enhancing tabular data synthesis. Frontiers in Big Data, 6, 2024. ISSN 2624-909X. doi: 10.3389/ fdata.2023.1296508. URL https://www.frontiersin.org/journals/big-data/ articles/10.3389/fdata.2023.1296508.

---

### Official Review · Reviewer_vt7B · 2024-11-03

**Soundness:** 3
**Presentation:** 3
**Contribution:** 3
**Rating:** 6
**Confidence:** 2

**Summary:**

This paper proposes a post-processing method to ensure that already generated tabular data satisfies conditions defined as a union of feasible sets specified by linear inequalities. The constraints include complex forms, such as disconnected and non-convex sets, which are rarely addressed in traditional constraint problems. The proposed method, a variation of Fourier-Motzkin, progressively reduces constraints to derive a broader feasible region. If the generated data fails to meet predefined constraints, the original data point is adjusted to the one with the minimum Euclidean distance within the relaxed constraint region.

**Strengths:**

When the feasible region is a disconnected, non-convex set, it is challenging to directly apply projection methods for post-adjusting generated data. This paper is significant because it demonstrates the existence of complex constraints in tabular data generation and proposes a solution to address this issue.

**Weaknesses:**

The method proposed in this paper for post-processing already generated data may require further theoretical support. It isn't easy to assert that finding the data point with the shortest Euclidean distance within the constraint-satisfying set is the most reasonable approach. For example, the Mahalanobis distance might be a more effective metric.

Furthermore, the paper does not compare information leakage and reproduction accuracy, essential performance metrics for generated data. It would be beneficial to demonstrate the trade-off between these two critical metrics in generative models. For example, it is helpful to report the Membership Inference Accuracy and Attribute Disclosure Risk of the considered models.

Another limitation of this model is that it requires ranking the variables included in the constraints to be satisfied for high-dimensional data. The proposed method seeks solutions within a feasible region larger than the region required to satisfy the constraints. Additionally, the constraint composed of the union of linear constraints covered in this paper could potentially be avoided during optimization using barrier methods. Although this approach differs from the method proposed in the paper, one could consider normalizing the model by using a penalty function to ensure the generated table model remains within the predefined feasible region. For instance, adding a log barrier function to the risk function can enforce the necessary constraints on the output of the VAE decoder. Alternatively, for computational stability, ReLU could be used instead of the log barrier function. This approach may serve as an alternative that fulfills both the objective function for model training and the given constraints, potentially bypassing the issue of defining metrics for post-processing generated data.

**Questions:**

1. In Lemma 3.1, what does " optimal " mean?

2. What is the meaning of "The CP resolution rule is sound"?

3. In Example 4, does the result depend on the ordering of variables?

4. Does the proposed method adjust the generated data to satisfy the given constraints in the problem, or does it use more relaxed constraints?

---

> ### Author Response · Authors · 2024-11-20
>
> Firstly, we would like to thank the reviewer for their appreciation of the novelty and relevance of our method. Further, we would like to thank them for their comments, which have given us interesting avenues to explore for future work.
>
> Below we provide point-by-point answers to all the questions.
>
> Q1. *The method proposed in this paper for post-processing already generated data may require further theoretical support. It isn't easy to assert that finding the data point with the shortest Euclidean distance within the constraint-satisfying set is the most reasonable approach. For example, the Mahalanobis distance might be a more effective metric.*
>
> A1. The Euclidean distance is computationally efficient, it is intuitive to understand and showed to work well in ensuring the satisfaction of the constraints while improving the Efficacy of the synthetic data. Additionally, using the Euclidean distance allowed us to follow the definition of optimality already given in [1], where only linear constraints are considered. This said, we totally agree with the reviewer that it is not the unique metric that can be used and it would be an exciting research question to study how the DRL layer would change when taking into account other metrics!
>
> Q2. *Furthermore, the paper does not compare information leakage and reproduction accuracy, essential performance metrics for generated data. It would be beneficial to demonstrate the trade-off between these two critical metrics in generative models. For example, it is helpful to report the Membership Inference Accuracy and Attribute Disclosure Risk of the considered models.*
>
> A2. As our method adds a layer (DRL) on top of existing generative models, we did not expect our method to have a big impact on the information leakage and reproduction accuracy. This is also confirmed by the results reported in the Table below, where the Identifiability Score (as defined in [2]) is shown. A 0-identifiability would represent a perfectly non-identifiable dataset and 1-identifiability would represent a perfectly identifiable dataset (hence the lower the better). Best results are in **bold**.
>
> | Model      | URL  | CCS  | LCLD | Heloc | House |
> |------------|------|------|------|-------|-------|
> | WGAN       | **0.24** | 0.32 | **0.23** | **0.21**  | 0.13  |
> | WGAN+DRL   | **0.24** | **0.17** | **0.23** | 0.27  | **0.08**  |
>
> As expected, DRL did not have a big impact on the identifiability score as WGAN+DRL got the same score as WGAN twice, performed better than WGAN twice and worse only once. We computed these scores for a single model in order to start the discussion as soon as possible. We will include them for all models upon publication if the paper gets accepted, and if possible, even during this discussion period.

---

> > ### Author Response · Authors · 2024-11-20
> >
> > Q3. *Another limitation of this model is that it requires ranking the variables included in the constraints to be satisfied for high-dimensional data. The proposed method seeks solutions within a feasible region larger than the region required to satisfy the constraints. Additionally, the constraint composed of the union of linear constraints covered in this paper could potentially be avoided during optimization using barrier methods. Although this approach differs from the method proposed in the paper, one could consider normalizing the model by using a penalty function to ensure the generated table model remains within the predefined feasible region. For instance, adding a log barrier function to the risk function can enforce the necessary constraints on the output of the VAE decoder. Alternatively, for computational stability, ReLU could be used instead of the log barrier function. This approach may serve as an alternative that fulfills both the objective function for model training and the given constraints, potentially bypassing the issue of defining metrics for post-processing generated data.*
> >
> > We will give a point-by-point answer to this comment.
> >
> > 1. We are not sure what the reviewer means by a region larger than the region required to satisfy the constraints. Did the reviewer mean that given a constraint $x < 3 \vee x > 10$ then it is sufficient to satisfy $x<3$ to satisfy the constraint? If that is the case, while ensuring that $x<3$ is enough to satisfy the constraint, it would also mean to only consider this subregion while training and then only generate datapoints with values lower than $3$ for the feature $x$. This would obviously lead to a degradation in the distribution of the generated datapoints. It is thus necessary to create a layer that is able to model the entirety of the space defined by the constraints.
> >
> > 2. Firstly, adding a barrier function to the loss function gives no guarantee that the model would always satisfy the constraints.  Secondly, we are not sure how barrier functions could work with constraints that define non-convex and even disconnected spaces. Indeed, barrier functions are primarily used with convex, continuous and smooth constraints. Moreover, even if we were able to adapt them to our constraints defining non-convex and disconnected spaces, our intuition is that the training of the VAE would become very unstable given the complex landscape of the loss function.
> >
> > 3. We are not sure what the reviewer means by simply adding a ReLU. Suppose we have 20 constraints over 10 variables. How could a ReLU be applied to ensure that the constraints are always satisfied?
> >
> >
> > Q4. *In Lemma 3.1, what does "optimal" mean?*
> >
> > We defined optimal at line 303 page 6 (of the original paper). DRL($\tilde x$) is said to be optimal w.r.t. $\tilde x$, $\Pi$ and the variable ordering if for each $i = 1, \ldots, D$ there does not exist a sample $\tilde x' \in \Omega(\Pi)$ such that $|\tilde x_i - \tilde x'_i| < |\tilde x_i - DRL(\tilde x)_i|$, and for all $j < i$,  $\tilde x'_j = DRL(\tilde x)_j$. This definition of optimality is taken from [1], and allows us to take into account the preferences about the set of features which we want to minimally change.
> >
> > Q5. *What is the meaning of "The CP resolution rule is sound"?*
> >
> > Soundness is a standard concept in logic, meaning that the conclusion is entailed by the premises, i.e., that any model (i.e., satisfying assignment) of the premises is also a model of the conclusion of the inference rule.
> >
> > Q6. *In Example 4, does the result depend on the ordering of variables?*
> >
> > Yes, the reviewer is correct. The constraints included in each $\Pi_i$ for $i=1, \ldots, D$ depend on the variable ordering.
> >
> > Q7. *Does the proposed method adjust the generated data to satisfy the given constraints in the problem, or does it use more relaxed constraints?*
> >
> > The proposed method adjusts the datapoints to satisfy the given constraints and each synthetic datapoint is thus guaranteed by-construction to be compliant with the given constraints. No constraint is ever relaxed.
> >
> > ***References:***
> >
> > [1] Stoian et al. How Realistic Is Your Synthetic Data? Constraining Deep Generative Models for Tabular Data, ICLR, 2024
> >
> > [2] Yoon et al. Anonymization Through Data Synthesis Using Generative Adversarial Networks (ADS-GAN), Journal of Biomedical and Health Informatics, 2020

---

> > > ### Comment · Reviewer_vt7B · 2024-11-24
> > >
> > > Thank you for your detailed responses. I lack sufficient background knowledge on this paper, so your answers have greatly helped me understand it. However, I have a few additional questions related to Q3 that I would like to confirm:
> > >
> > > 1. Does the DRL map proposed by the authors aim to find a new generated data point closest (in terms of Euclidean distance) to the set of constraints, starting from a generated data point that does not satisfy the given constraints?
> > >
> > > 2. Is the feasible region related to the DRL map in the paper expressed as a union of sets formed by the intersection of half-spaces?
> > >
> > > 3. Let me focus on feasible regions related to the constraints that are disconnected sets. Among the constraints mentioned in question 2 above, are the disconnected sets represented as a union of disconnected convex sets?
> > >
> > > 4. Is finding the newly generated data point through the DRL map the same as finding the point with the shortest distance to each disconnected convex set defined in question 3 above?

---

> > > > ### Author Response · Authors · 2024-11-24
> > > >
> > > > We thank the reviewer very much for their response and we are happy to provide any additional clarification.
> > > >
> > > > Q1. *Does the DRL map proposed by the authors aim to find a new generated data point closest (in terms of Euclidean distance) to the set of constraints, starting from a generated data point that does not satisfy the given constraints?*
> > > >
> > > > A1. If a single variable appears in each of the constraints, then yes, given a datapoint that does not satisfy the given constraints, DRL maps it to the point in the solution space defined by the constraints that is closest (in terms of Euclidean distance). More generally, if we consider a non-compliant sample $\tilde x$ with $D$ features, and the ordering $x_1;x_2;\ldots;x_D$, we return $DRL(\tilde x )$ such that, for every feature $x_i$, $DRL(\tilde x)_i$ is at minimum Euclidean distance from $x_i$ given the values of $DRL(\tilde x)_j$ for $j<i$.
> > > >
> > > > Q2. *Is the feasible region related to the DRL map in the paper expressed as a union of sets formed by the intersection of half-spaces?*
> > > >
> > > > A2. Each constraint is expressed as a union of half-spaces. The set of constraints taken as input by DRL is expressed as the intersection of the spaces defined by the constraints. Every set of constraints $\Pi_i$ defined by DRL is expressed again as an intersection over unions of half-spaces.
> > > >
> > > > Q3. *Let me focus on feasible regions related to the constraints that are disconnected sets. Among the constraints mentioned in question 2 above, are the disconnected sets represented as a union of disconnected convex sets?*
> > > >
> > > > A3. A convex space is always connected and constraints might be the union of non-disjoint spaces. For example, the constraint $(x_2 \ge 0) \vee (x_2 \le x_1)$  defines a non-convex space that is connected and is the union of the two convex spaces, i.e., $(x_2 \ge 0)$ and $(x_2 \le x_1)$. However, at inference time, when deciding the value of a feature (e.g., $x_2$) given the values of the preceding features (e.g., $x_1 = -5$), the feasible region (e.g., for $x_2$) can be the union of disjoint convex sets.
> > > >
> > > > Q4. *Is finding the newly generated data point through the DRL map the same as finding the point with the shortest distance to each disconnected convex set defined in question 3 above?*
> > > >
> > > > A4. Please refer to our answer to Q1 w.r.t. what is the goal of DRL and to our answer to Q3 w.r.t. the space that our constraints define. The intuition is correct when considering what our layer does feature-wise at inference time. Indeed, for each feature DRL defines a set of convex and disjoint subsets of the solution space and then our goal is to map the value of the considered feature, if non-compliant, to the closest feasible subset.

---

> > > > > ### Comment · Reviewer_vt7B · 2024-11-28
> > > > >
> > > > > Thank you for your response. I have adjusted my score.

---

### Official Review · Reviewer_TebA · 2024-11-03

**Soundness:** 3
**Presentation:** 3
**Contribution:** 2
**Rating:** 6
**Confidence:** 2

**Summary:**

This paper introduces a novel Disjunctive Refinement Layer (DRL) for Deep Generative Models (DGMs), aiming to generate synthetic tabular data that 1) the sample space of the model distribution is aligned with what is stated in the background knowledge. 2) The model distribution approximates the ground truth distribution. This paper extends constraint enforcement in data generation by leveraging Quantifier-Free Linear Real Arithmetic (QFLRA) formulas, enabling more complicated (e.g. non-convex constraints). The theorem proves the proposed DRL fully satisfies the constraints and is optimal in Euclidean distance.

**Strengths:**

1. The use of QFLRA constraints in DRL is a substantial advancement, enabling the generation of data that complies with complex, realistic conditions. This advancement is especially relevant for applications requiring high data fidelity, such as healthcare or finance.

2. Theory analysis proves the DRL will transform the generated data to fully satisfy the constraints while attaining optimal Euclidean distance to the original sample.

3. The paper includes extensive testing across five generative models and datasets, showing consistent improvements in constraint satisfaction and downstream model performance.

**Weaknesses:**

The main method section (Sec 3) is a little bit dense and hard to read. I think providing a pseudo code is helpful here.

**Questions:**

1. What is the computation complexity of DRL?

2. What are the constraints for different datasets? Are they manually chosen by authors or are included in the original datasets?

2. Why does satisfying the constraints help improve machine learning efficiency? Is that because we also transform the test dataset so that they satisfy the constraints?

---

> ### Author Response · Authors · 2024-11-20
>
> We thank the reviewer their appreciation of the novelty, relevance and soundness of our method and experimental analysis.
>
>  Below we give a point-by-point answer to all the questions. In the new version of the PDF we have highlighted all the new text related to their review in *light blue*.
>
> Q1. *The main method section (Sec 3) is a little bit dense and hard to read. I think providing a pseudo code is helpful here.*
>
> A1. We thank the reviewer for the suggestion of adding the pseudo-code. We added it in Section 3 and we believe it provides a nice bird-eye view of the method.
>
> Q2. *What is the computational complexity of DRL?*
>
> A2. The DLR compilation step (which only happens once before training) creates a number of constraints which critically depends on the number of non-convex constraints in $\Pi$ or derived from $\Pi$ with CP-resolution. In the worst-case, happening when for each variable $x_i$ the size of $\Pi^\pm_i$ is proportional to the size of $\Pi$, DLR can be super-exponential in the size of $\Pi$. However, we never experienced an exponential blow-up and hence memory problems in our experiments.  It is also worth pointing out that once the DRL is built, starting from a sample $\tilde x$, the computation of DRL($\tilde x$):
>    1. takes linear time in the number of levels of DRL, which is at most equal to the number of variables occurring in $\Pi$, and
> 2. at each level $i$, the operations performed to compute DRL($\tilde x_i$)  are standard (e.g., max, min, sum etc,) each of which has been heavily optmised by pytorch to run in parallel on GPUs.
>
> In practice, in our experiments, we never experienced a significant degradation of the performance produced by DRL, as shown in Table 5.
>
> Q3. *What are the constraints for different datasets? Are they manually chosen by authors or are included in the original datasets?*
>
> A3. The URL, LCLD and Heloc datasets were already annotated with such constraints. Regarding CCS and House, we manually annotated the constraints using our background knowledge about the problem, then we checked whether the data were compliant with our constraints and finally we retained only those constraints that were satisfied by **all** the datapoints. Simple examples of these constraints (from CCS) state simple facts like “if the feature capturing the number of cigarettes packs per day is greater than 0 then the feature capturing whether the patient smokes or not should be equal to 1” or “the feature representing the age should always be higher than the age at the first intercourse”.
>
> Q4. *Why does satisfying the constraints help improve machine learning efficiency? Is that because we also transform the test dataset so that they satisfy the constraints?*
>
> A4. To explain why we get better machine learning efficiency, consider the House dataset, for which the target feature is the house price and we visualise the violation areas in Figure 4 (page 7) and 5 (page 23). For this dataset, we have among others two constraints simply stating that if a house is located in zipcodes 98004 or 98005 then the house price should be above 400,000USD and if the house is located in zipcodes 98006, 98007 or 98008 then the house price should be above 200,000USD. As we can see from the Figures all the standard generative models (TVAE, WGAN, TableGAN and GOGGLE) often violate the constraint, thus creating datapoints where for example a house located at zipcode 980004 is worth only 100,000USD. Now suppose we train a classifier on the datapoints generated with TVAE. Given these synthetic datapoints, the classifier will now wrongly learn that a house in zipcode 98004 can be worth well below 400,000USD, hence leading to worse predictions and higher Mean Square Error. This example clearly shows why by adding DRL we get higher machine learning efficacy. Finally, note that the constraints express known relationships between the datapoints that hold for all of them. Hence, no datapoint in the test set has ever been modified.

---

### Official Review · Reviewer_oZm4 · 2024-11-06

**Soundness:** 3
**Presentation:** 3
**Contribution:** 3
**Rating:** 8
**Confidence:** 2

**Summary:**

The authors of this paper study the problem of generating tabular data where complex constraints expressed by QFLRA should be satisfied. The authors propose a neuro-symbolic AI layer to tackle this problem and prove its effectiveness in respecting the constraints. Experimental results on several datasets clearly demonstrate the advantages of the proposed model over the baselines.

**Strengths:**

S1. The problem of satisfying QFLRA in building tabular data generative models is new.
S2. The proposed technique can be integrated into any deep model.
S3. The proposed method is sound as the authors prove its correctness.

**Weaknesses:**

W1. The authors mainly focus on proving that their proposed Neuro-symbolic AI layer can respect the complex constraints expressed by QFLRA. What about the impact of this new layer on the learned distribution? Is it possible that such a new layer may harm the learned distribution for satisfying the constraints?

W2. It seems to me that the constraints of QFLRA may be very complex and even deciding if a QFLRA is satisfiable is a difficult problem. What would happen to the proposed model if the input QFLRA is not satisfiable?

W3. In the experiments, the authors adopt some simple QFLRA constraints, where the feasible region has a couple of disconnected parts. For these relatively simple QFLRA, can we first identify each disconnected subregion, then use traditional models to learn the distribution over each subregion and exploit a mixture model to combine distributions over all subregions to have an overall generative model?

**Questions:**

Please refer to W1-W3.

---

> ### Author Response · Authors · 2024-11-20
>
> We thank the reviewer for their appreciation of (i) the novelty of the problem we are trying to solve and its hardness, (ii) the generality of our solution and (iii) the soundness of our method. In the new version of the PDF we have highlighted all the new text related to their review in *orange*.
>
> Below we give a point-by-point answer to all the questions.
>
> Q1. *The authors mainly focus on proving that their proposed Neuro-symbolic AI layer can respect the complex constraints expressed by QFLRA. What about the impact of this new layer on the learned distribution? Is it possible that such a new layer may harm the learned distribution for satisfying the constraints?*
>
> A1. We thank the reviewer for raising this point. Please note that our paper has two focuses: (i) providing guarantees that our layer always satisfies the constraints and (ii) showing that more realistic synthetic datapoints (i.e., compliant with the background knowledge) also have better machine learning efficacy. The fact that we consistently improve the machine learning efficacy of the synthetic datasets is already an indicator that the new layer does not harm the learnt distribution, and actually makes it even more similar to the real distribution. To further prove this point, we also perform for each continuous feature the Kolmogorov-Smirnov test between the real datapoints and the samples from (i) WGAN and (ii) WGAN+DRL. The test measures the probability that two sets of samples come from the same distribution, thus the closer the result is to 1 the better the synthetic datapoints capture the real distribution. Below we report the results:
>
> | Model      | URL  | CCS  | LCLD | Heloc | House |
> |------------|------|------|------|-------|-------|
> | WGAN       | 0.59 | 0.39 | 0.70 | 0.74  | 0.61  |
> | WGAN+DRL   | 0.88 | 0.58 | 0.75 | 0.82  | 0.73  |
>
> We computed these results for a single model in order to start the discussion as soon as possible. We will include them for all models upon publication if the paper gets accepted, and if possible, even during this discussion period.
>
>
> Q2. *It seems to me that the constraints of QFLRA may be very complex and even deciding if a QFLRA is satisfiable is a difficult problem. What would happen to the proposed model if the input QFLRA is not satisfiable?*
>
> A2. We thank the reviewer for this point, which allowed us to make our paper clearer. As written in the paper, CP resolution is refutationally complete: this means that if the set of QFLRA constraints is unsatisfiable then it is possible to derive a disjunction of linear inequalities in which each inequality (3) has $w_i = 0$ for $i = 1, \ldots , D$ and $b < 0$. At the end of the layer construction, we are able to automatically detect $\Pi$ unsatisfiability, returning a corresponding UNSAT_FLAG value in such a case. We have added this last phrase in the paper to make it even clearer. (page 6)
>
>
> Q3. *In the experiments, the authors adopt some simple QFLRA constraints, where the feasible region has a couple of disconnected parts. For these relatively simple QFLRA, can we first identify each disconnected subregion, then use traditional models to learn the distribution over each subregion and exploit a mixture model to combine distributions over all subregions to have an overall generative model?*
>
> A3. The goal of our method is to provide a general solution that works for any set of constraints. Additionally, it is not clear how to identify the disconnected components and then train in an efficient way a different unconstrained model that is guaranteed to be compliant with the constraints. If we understand correctly what the reviewer has in mind, this would amount to apriori $(i)$ compute the DNF of the input set of constraints. $(ii)$ make the disjuncts pairwise inconsistent (e.g., going from $(A \vee B)$ to $(A \wedge \neg B) \vee B)$), and then $(iii)$ consider the subspace corresponding to each disjunct (conjunction of constraints), which can be exponentially many.  Even supposing we can find the disconnected components and train a different traditional model for each one of them, it is still not clear how to ensure that the constraints are not violated by unconstrained models which violate constraints having convex and connected solution spaces (as demonstrated by the fact that the unconstrained models often violate even the constraints expressed as linear inequalities)

---

> > ### Comment · Reviewer_oZm4 · 2024-11-22
> >
> > Thank you for the clarification. I do not have further questions.

---

### Author Response · Authors · 2024-11-25

We are very thankful to the reviewers for their feedback, which has made our paper more comprehensive.

We also thank them for appreciating the clarity of our paper (oZm4), the hardness of the problem handled (oZm4,vt7B,KTDt), the extent of our experimental analysis (TebA), and the soundness of our approach (oZm4,KTDt), together with its novelty (oZm4,KTDt), its generality (oZm4) and its relevance (TebA,vt7B).

As the discussion phase comes to a close, we kindly ask the reviewers if there are any additional questions. If we resolved all their comments, we would be very grateful if the reviewers would consider raising their score.



Thanks again for all the efforts,

The authors

---

### Author Response · Authors · 2024-12-02

Dear reviewers,

We would like to express our gratitude once again for your thoughtful feedback which has led to a more comprehensive version of our paper.

As the discussion period deadline approaches, we would like to kindly ask if there are any additional questions. If not, we would greatly appreciate it if you would consider adjusting the scores accordingly.

Thank you for your time and effort!


Best,

The authors

---

### Meta-Review · Area_Chair_8kMv · 2024-12-12

**Metareview:**

This paper introduces a new neuro-symbolic layer for DGMs to improve tabular data generation given background knowledge. It allows the specification of non-convex constraints and as a result the DGM only generates points satisfying those constraints, which can even improve the distributional similarity of generated data to test data.

The reviewers brought up several concerns with the paper, but these were largely clarified during the discussion period with efforts made by the authors to provide new experimental results as requested. None of the remaining concerns are major, and can be worked on by the authors before the camera-ready submission. I am recommending acceptance as a Poster.

**Additional Comments On Reviewer Discussion:**

The main points of improvement raised by reviewers were: lack of complexity analysis or empirical results on generation speed; lack of consideration of alternative options and details (e.g. choice of Euclidean distance, training a model for each disjoint region); general clarifications on math and experimental detail. The authors addressed the points with additional experiments that the reviewers were satisfied with.

---

### Decision · Program_Chairs · 2025-01-22

Accept (Poster)